Corrected: Publisher correction

# NRG1 type I dependent autoparacrine stimulation of Schwann cells in onion bulbs of peripheral neuropathies

Robert Fledrich [1,2], Dagmar Akkermann[2,3], Vlad Schütza[2,3], Tamer A. Abdelaal [2,3], Doris Hermes [2,4], Erik Schäffner[2,3], M. Clara Soto-Bernardini[2,5], Tilmann Götze[2,6], Axel Klink[2], Kathrin Kusch[2], Martin Krueger[1], Theresa Kungl[2], Clara Frydrychowicz[3], Wiebke Möbius [2,7], Wolfgang Brück[8], Wolf C. Mueller[3], Ingo Bechmann[1], Michael W. Sereda[2,4], Markus H. Schwab [2,6,9], Klaus-Armin Nave[2] & Ruth M. Stassart[2,3]

In contrast to acute peripheral nerve injury, the molecular response of Schwann cells in chronic neuropathies remains poorly understood. Onion bulb structures are a pathological hallmark of demyelinating neuropathies, but the nature of these formations is unknown. Here, we show that Schwann cells induce the expression of Neuregulin-1 type I (NRG1-I), a paracrine growth factor, in various chronic demyelinating diseases. Genetic disruption of Schwann cell-derived NRG1 signalling in a mouse model of Charcot-Marie-Tooth Disease 1A (CMT1A), suppresses hypermyelination and the formation of onion bulbs. Transgenic overexpression of NRG1-I in Schwann cells on a wildtype background is sufficient to mediate an interaction between Schwann cells via an ErbB2 receptor-MEK/ERK signaling axis, which causes onion bulb formations and results in a peripheral neuropathy reminiscent of CMT1A. We suggest that diseased Schwann cells mount a regeneration program that is beneficial in acute nerve injury, but that overstimulation of Schwann cells in chronic neuropathies is detrimental.

[1] Institute of Anatomy, University of Leipzig, Liebigstr. 13, 04103 Leipzig, Germany. [2] Department of Neurogenetics, Max-Planck-Institute of Experimental Medicine, Hermann-Rein-Str. 3, 37075 Göttingen, Germany. [3] Department of Neuropathology, University Hospital Leipzig, Liebigstr. 26, 04103 Leipzig, Germany. [4] Department of Clinical Neurophysiology, University Medical Center Göttingen, Robert-Koch-Str. 40, 37075 Göttingen, Germany. [5] Center for Research in Biotechnology (CIB), Costa Rican Institute of Technology (TEC), Cartago, Costa Rica. [6] Department of Cellular Neurophysiology, Hanover Medical School, Carl-Neuberg-Str. 1, 30625 Hanover, Germany. [7] Center Nanoscale Microscopy and Molecular Physiology of the Brain (CNMPB), Göttingen, Germany. [8] Institute of Neuropathology, University Medical Center Göttingen, Robert-Koch-Str. 40, 37075 Göttingen, Germany. [9] Center for Systems Neuroscience (ZSN), Bünteweg 2, 30559 Hanover, Germany. These authors contributed equally: Robert Fledrich, Dagmar Akkermann. Correspondence and requests for materials should be addressed to R.F. (email: robert.fledrich@medizin.uni-leipzig.de) or to M.H.S. (email: schwab.markus@mh-hannover.de) or to K.-A.N. (email: nave@em.mpg.de) or to R.M.S. (email: ruth.stassart@medizin.uni-leipzig.de)

Schwann cells ensheath peripheral nerve axons with myelin membranes that provide electrical insulation for rapid impulse conduction[1]. Genetic defects that impair Schwann cell function underlie a heterogeneous group of demyelinating neuropathies, collectively referred to as Charcot–Marie–Tooth (CMT) disease, which affects approximately 1 in 2500 humans[2]. The most common subtype, CMT1A, is caused by an interstitial duplication on chromosome 17, resulting in overexpression of the gene encoding the peripheral myelin protein of 22 kDa (PMP22), a small hydrophobic protein of unknown function and an integral constituent of peripheral nerve myelin[3–5]. Patients affected by CMT1A suffer from a slowly progressive, distally pronounced muscle weakness and sensory deficits[6]. Although patients usually seek medical advice in young adulthood, CMT1A manifests already during childhood by mild walking disabilities and a pronounced slowing of nerve conduction velocity (NCV), suggesting malfunction of the myelin sheath[7]. Indeed, peripheral nerves of CMT1A patients are characterized by developmental dysmyelination, including hypermyelination of small to mid-caliber axons and reduced internodal length[8,9]. Along with disease progression, demyelination and axonal loss become apparent, in addition to numerous onion bulb formations. The latter are concentrically aligned supernumerary Schwann cell processes that enwrap an inner axon–Schwann cell unit and represent a key histological disease hallmark of CMT1A disease[10–12]. Of note, onion bulb structures have long been used as a cardinal diagnostic criterion for demyelinating neuropathies in sural nerve biopsies from human patients. Onion bulb formations have been hypothesized to derive from displaced surviving Schwann cells that are generated during repetitive cycles of demyelination and remyelination[13–15]. However, the (glial) pathomechanisms that contribute to this common pathway of disease expression remain poorly understood. Within the present manuscript, we hence aimed at identifying the molecular mechanisms that cause onion bulb formations in peripheral neuropathies.

Recently, a dysdifferentiated phenotype similar to the dedifferentiation state of Schwann cells after acute nerve injury has been observed in Schwann cells of CMT1A disease[16,17], suggesting that diseased Schwann cells in acute and chronic peripheral nerve diseases may have been exposed to common pathomechanisms. After acute nerve injury, Schwann cells revert from mature myelinating cells to proliferating immature cells, in a process referred to as dedifferentiation or transdifferentiation[18]. Although the responsible upstream mechanisms remain elusive, the process of dedifferentiation is controlled by the re-activation of mitogen-activated extracellular signal-regulated kinase (Mek)/extracellular signal–regulated kinase (Erk) signaling and a network of transcriptional regulators in adult Schwann cells[19], with a major role for the transcription factor cJUN[20]. Subsequently, dedifferentiated Schwann cells align in the bands of Büngner and finally redifferentiate and remyelinate regenerated axons[18].

During peripheral nerve development, Schwann cell differentiation and myelination critically depend on axon-derived growth factors, namely Neuregulin-1 (NRG1)[21]. NRG1 belongs to a family of transmembrane and secreted epidermal growth factor (EGF)-like growth factors, which exist in various isoforms and share an EGF-like domain that is sufficient and required for the activation of ErbB receptor tyrosine kinases[21–23]. When expressed on the axonal surface, the transmembrane NRG1 type III isoform controls virtually all steps of Schwann cell development and ultimately regulates myelin sheath thickness[21,23,24]. High levels of NRG1 type II and type III, however, have been demonstrated to induce demyelination and transgenic overexpression of NRG1 type II in Schwann cells leads to tumorigenesis preceded by a hypertrophic onion bulb pathology[25,26]. Of note, NRG1 expression on the axonal surface is barely detectable in adulthood and dispensable for the maintenance of adult nerve functions[27,28]. However, Wallerian degeneration of nerve fibers induces a de novo expression of the soluble Neuregulin-1 type 1 (NRG1-I) isoform in Schwann cells, a timely restricted signal that supports nerve repair and remyelination after acute nerve injury[29].

Here we demonstrate that Schwann cells in chronic demyelinating neuropathies specifically induce expression of the paracrine NRG1-I isoform, which is required for disease pathogenesis in a CMT1A mouse model. Conditional *Nrg1* ablation in Schwann cells reduces major pathological disease hallmarks, including dysmyelination and onion bulb formation, and strongly ameliorates the clinical disease phenotype of adult CMT1A mice. Schwann cell-specific overexpression of soluble NRG1-I in transgenic mice reveals that onion bulb structures evolve as a consequence of glial NRG1-I-mediated survival, attraction, and alignment of supernumerary dedifferentiated Schwann cells. We suggest a model according to which Schwann cells in peripheral nerve diseases mount a common NRG1-I-mediated regeneration program, which is beneficial after acute nerve injury but turns into a detrimental persistent response in chronic peripheral neuropathies, such as CMT1A.

## Results

**Glial NRG1-I expression is induced in demyelinating neuropathies.** Glial NRG1-I expression is upregulated following acute nerve injury[29]. To examine soluble NRG1-I expression in chronic peripheral nerve diseases, we performed an expression analysis in various rodent models for peripheral neuropathies at adult disease stages. Notably, we found *Nrg1-I* transcripts to be upregulated in chronically injured peripheral nerves, with the highest expression in demyelinating forms of peripheral neuropathies (Fig. 1a). Importantly, increased *NRG1-I* expression was also a feature of human sural nerve biopsies characterized by a demyelinating histopathology (Fig. 1b, c), independent of the underlying primary disease cause. Of note, *NRG1-I* transcriptional levels varied among individual patients and a comparison to the histopathological phenotype revealed a significant positive correlation of *NRG1-I* expression with the number of onion bulb formations in sural nerve biopsies (Fig. 1c). Could the induction of glial NRG1-I underlie onion bulb formation in demyelinating neuropathies?

In order to address whether NRG1-I contributes to this final common pathway of disease expression, we first investigated the time course of *Nrg1-I* mRNA expression in rodent models of CMT1A disease (*Tg(Pmp22)Kan*, termed CMT1A rat[30], and *Tg(PMP22)C61Clh*, termed CMT1A mouse[31]). Here *Nrg1-I* mRNA expression was strongly upregulated during early postnatal development (P6–18) in the CMT1A rat and mouse models and correlated with the early postnatal myelination defect in CMT1A disease[16]. In contrast, *Nrg1-II* and *Nrg1-III* transcription was unaltered in both models (Fig. 1d, Supplementary Fig. 1a). A further increase in expression levels was observed with age and disease progression and included *Nrg1-I* transcripts encoding both the alpha- and beta-type EGF-like domain (Fig. 1d, Supplementary Fig. 1a). In line with RNA data, the C-terminal domain of NRG1 protein was readily detectable in nerve lysates from 3-month-old CMT1A rats but not in wild type (Fig. 1e). This is consistent with preferential axonal transport of the N-terminal portion of NRG1 (lacking the C-terminus) during myelination[28] and low axonal NRG1 expression in adulthood. Accordingly, the NRG1 C-terminus localized to Schwann cells (but not axons) in CMT1A mouse nerve sections but was not detectable in wild type (Fig. 1f).

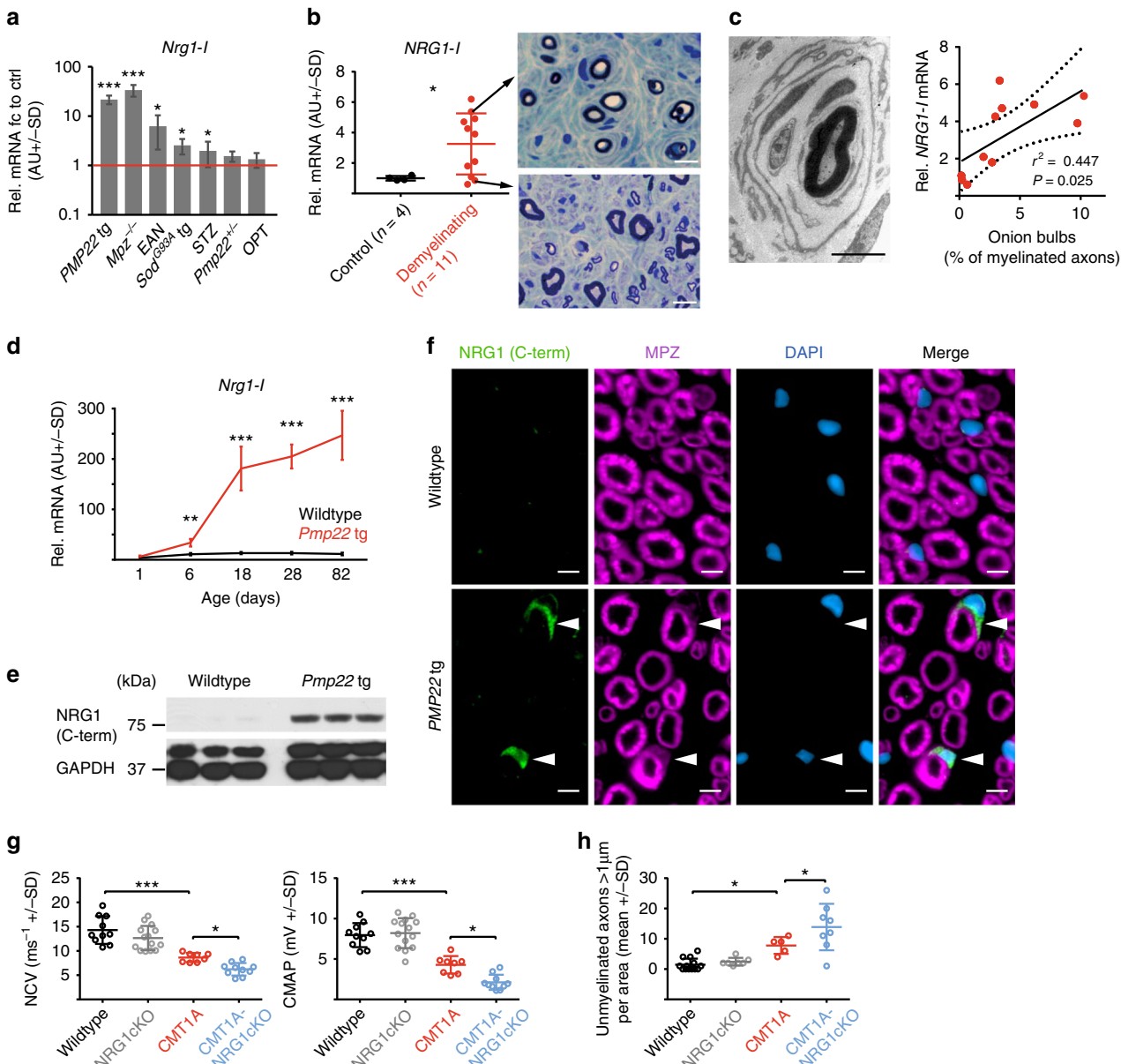

**Fig. 1** Neuregulin-1 type 1 (NRG1-I) expression by Schwann cells in various demyelinating neuropathies. **a** *Nrg1-I* mRNA expression in sciatic nerves of different rodent models for peripheral neuropathies. *Nrg1-I* mRNA induction is most pronounced in demyelinating models (*PMP22* tg mice, CMT1A model; *Mpz*−/− mice, model for congenital hypomyelination; EAN rats, experimental autoimmune neuritis) compared to predominant axonal neuropathies (*SOD*$^{G93A}$ tg mice, model for amyotrophic lateral sclerosis; Streptozotocin (STZ)-treated mice, diabetic neuropathy model; *Pmp22*$^{+/−}$ mice, model for hereditary neuropathy with liability to pressure palsies; Oxaliplatin (OPT)-treated rats, toxic neuropathy model). Sciatic nerves were collected from adult (age 3–4 months) experimental rodents and age-matched controls, respectively (n = 5 per group). Shown is *Nrg1-I* mRNA expression relative to control level (=1, red line). (Student's *T* test). **b** *NRG1-I* mRNA expression (standardized to Schwann cell marker *S100*) is increased in sural nerve biopsies with a demyelinating histopathology (n = 11) compared to controls (n = 4, Student's *T* test, left panel). Semithin cross-sections of representative human sural nerve biopsies with high and low *NRG1-I* expression are shown in right panels (scale bar 10 μm). **c** Pearson correlation analysis (right panel) of sural nerve *NRG1-I* mRNA expression and onion bulb frequency reveals a significant positive correlation between *NRG1-I* expression and the number of onion bulbs (data from **b**). An electron micrograph exemplarily depicts an onion bulb (scale bar 5 μm). **d** In CMT1A rats, *Nrg1-I* mRNA expression in sciatic nerves increases with age and disease progression (n = 4–5 per group, multiple *T* tests). **e** Western blot analysis of NRG1 protein (C-terminal antibody) in sciatic nerve lysates of 3-month-old CMT1A rats compared to wild-type animals (n = 3 per group, GAPDH as loading control; for full blots, see Supplementary Figure 5). **f** Immunohistochemistry on sciatic nerve cross-sections from 3-month-old wild-type (upper panels) and CMT1A mice (lower panels) showing NRG1 expression (green, C terminal SC348 antibody) only in Schwann cells of CMT1A mice. Myelin was stained with MPZ (magenta) and nuclei were with DAPI (blue) (scale bar 2.5 μm). **g** Quantification of nerve conduction velocity (NCV, left panel) and compound muscle action potential (CMAP) amplitudes (right panel) in P18 mice (wild type n = 10, NRG1cKO n = 14, CMT1A n = 8, CMT1A-NRG1cKO n = 10, one-way analysis of variance (ANOVA) and Tukey's post test). **h** Quantification of the number of unmyelinated axons (>1 μm) per area (15,740 μm$^2$) on electron microscopic level of P18 mice (wild type n = 12, NRG1cKO n = 7, CMT1A n = 5, CMT1A-NRG1cKO n = 8, one-way ANOVA and Tukey's post test). Source data are provided as a source data file. All respective *p* values are depicted as a range of significance with *$p < 0.05$; **$p < 0.01$; and ***$p < 0.001$

**Ablation of glial NRG1-I ameliorates the clinical phenotype of CMT1A mice**. To directly address a role of Schwann cell-derived NRG1 in CMT1A models, we ablated *Nrg1* in Schwann cells from CMT1A mice by breeding to conditional *Dhh*$^{Cre}$::*Nrg1-flox* mutants (FVB(Cg)-Tg(Dhh-cre)1Mejr/J, [*Dhh*$^{Cre}$][32] and *Nrg1*$^{tm3Cbm}$ [*Nrg1*$^{fl/fl}$][33], NRG1cKO mice), termed CMT1A-NRG1cKO mice. Analysis of CMT1A-NRG1cKO mice at P18 revealed a deterioration of the disease phenotype, with an impaired NCV and a lowered compound muscle action potential (CMAP) (Fig. 1g). In line, the number of unmyelinated axons was increased in sciatic nerves of CMT1A-NRG1cKO mutants at P18 (Fig. 1h), suggesting that Schwann cell-derived NRG1 promotes myelination in CMT1A mice during postnatal stages.

Importantly, supportive functions served by glial NRG1 in CMT1A Schwann cells were restricted to postnatal development, as adult CMT1A-NRG1cKO mice were not increasingly clinically impaired, but instead showed a strong amelioration of their clinical phenotype (Fig. 2a, b). Remarkably, motor functions of adult CMT1A-NRG1cKO animals even reached wild-type levels when quantified by rotarod and foot print analyses at 3 months of age (Fig. 2a, b). In line, electrophysiological recordings in adult CMT1A-NRG1cKO mutants revealed an improved NCV compared to single CMT1A mice (Fig. 2c, d). While the CMAP was not improved when compared to single CMT1A mice (Fig. 2c, e), the aberrantly increased temporal dispersion of CMAP amplitudes in CMT1A mice[6], reflecting asynchronous conducting nerve fibers, was ameliorated in CMT1A-NRG1cKO mutants, further indicating improved nerve function (Fig. 2c, f). Does the beneficial effect of NRG1 ablation persist during disease progression in CMT1A animals? Indeed, when we analyzed CMT1A-NRG1cKO mice at the age of 11 months, we still observed a significant amelioration of the clinical phenotype in CMT1A-NRG1cKO mice (Supplementary Fig. 1b), which was accompanied by improved NCV and CMAP (Supplementary Fig. 1c).

**Schwann cell NRG1 drives histopathological disease hallmarks in CMT1A**. We next asked whether ablation of glial NRG1 also improves deficits in myelin architecture in CMT1A nerves[8,11,16,34]. Analysis at 4 and 11 months of age revealed no changes in the total number of myelinated axons per peripheral nerve in CMT1A-NRG1cKO mutants, although we noted an improved axonal caliber distribution in 11-month-old mice (Fig. 2g, Supplementary Fig. 1d–f). Likewise, Schwann cell-specific loss of NRG1 had no normalizing effect on shortened internodal length in CMT1A mice (Supplementary Fig. 2a). However, the pathological hypermyelination of small caliber axons that is characteristic of CMT1A nerves was ameliorated in CMT1A-NRG1cKO mutants as assessed by *g*-ratio analysis (Fig. 2h, i; Supplementary Fig. 2b, c), though *Nrg1* ablation did not result in an overall reduction of myelin sheath thickness (Fig. 2h, i, Supplementary Fig. 2b, c).

Moreover, next to hypermyelination, we observed a marked reduction in the number of onion bulbs in CMT1A-NRG1cKO animals compared to single CMT1A mice at the age of 4 months (Fig. 2j, k), which remained strongly decreased throughout disease progression, i.e., in 11-month-old animals (Supplementary Fig. 2d). In summary, we conclude that glial NRG1 functions as a key driver of histopathological hallmarks including hypermyelination and onion bulb formations in CMT1A disease, which prompted us to test whether chronically increased Schwann cell-derived NRG1 signaling is sufficient to induce aspects of CMT1A pathology in vivo.

**NRG1-I overexpression in Schwann cells induces a peripheral neuropathy**. For this purpose, we generated two conditional transgenic mouse lines that permit Cre-mediated overexpression of *Nrg1-I-β1* and *Nrg1-IIIβ1* isoforms in Schwann cells, respectively (Fig. 3a, Soto-Bernardini, Götze and Schwab, unpublished). At the transcriptional level, *Nrg1-I and Nrg1-III mRNA* overexpression was 10–15-fold in *Dhh*$^{Cre}$::*Nrg1-I*$^{stop-flox}$ (termed NRG1-I OE mice) and *Nrg1-III*$^{stop-flox}$ mice, respectively, comparable in extent to the endogenous upregulation of *Nrg1-I* mRNA in CMT1A mice (Supplementary Fig. 2e, Supplementary Fig. 1a). Western blot analysis of sciatic nerves from NRG1-I OE mice revealed the expression of a 120 kDa full-length NRG1 protein as well as a 55 kDa N-terminal (EGF-like-domain containing) fragment, consistent with proteolytic cleavage of full-length NRG1-I in the juxtamembrane region[35] (Fig. 3b). This 55 kDa NRG1-I protein accumulated in CMT1A transgenic Schwann cells and was lost upon glial NRG1 ablation in CMT1A-NRG1cKO mice (Fig. 3b). Importantly, *Pmp22* expression itself was not affected by NRG1-I overexpression (Supplementary Fig. 2f), in line with the notion that *Pmp22* overexpression mediates pathomechanisms upstream of glial NRG1.

Phenotype assessment identified a clinical impairment in adult NRG1-I OE mice, with a reduced motor performance as determined by rotarod and grip strength analysis (Fig. 3c, d). Electrophysiological recordings showed reduced NCV but no alterations in CMAP (Fig. 3e), consistent with unaltered axonal numbers in NRG1-I OE mice (Fig. 3f). Of note, internodal length was not shortened but slightly elongated in NRG1-I OE mice compared to wild type (Supplementary Fig. 2a). However, we found NRG1-I OE mice to share other key histopathological features with CMT1A mice. In particular, NRG1-I overexpression in Schwann cells induced hypermyelination of small to mid-caliber axons, without affecting myelin compaction or myelin sheath thickness of larger caliber axons (Fig. 3g, h). Likewise, the axonal diameter distribution in NRG1-I OE animals remained unaltered (Supplementary Fig. 2g). Moreover, we observed an increased process extension of those Schwann cells that were not associated with axons during the ultrastructural analysis of NRG1-I OE mice (Fig. 3i). Indeed, when quantified on sciatic nerve cross-sections NRG1-I OE mice consistently showed occasional onion bulb-like structures with extra Schwann cells forming concentric processes around a central Schwann cell-axon unit (reminiscent of CMT1A pathology), something we never observed in controls (Fig. 3j, k).

As hypermyelination of small to mid-caliber axons is also a feature of transgenic mice overexpressing NRG1 type III in neurons (C57BL/6-Tg(Thy1-Nrg1*III)1Kan+/−, (*Nrg1-III* tg)[24]) and is associated with a reduced NCV and an impaired motor phenotype without axonal degeneration or altered CMAP (Supplementary Fig. 2h–j), we hence asked whether glial overexpression of NRG1-III would result in a similar histopathological phenotype as observed in NRG1-I OE mice. However, in stark contrast to NRG1-I OE mice, glial-specific overexpression of NRG1-III in *Dhh*$^{Cre}$::*Nrg1-III*$^{stop-flox}$ mice (see Fig. 3a) resulted in dramatic phenotypic changes, including postnatal growth retardation (Supplementary Fig. 2k), abnormal myelination of entire Remak bundles, and massive Schwann cell hyperplasia, indicative of peripheral nerve sheath tumorigenesis (Supplementary Fig. 2l–n).

In summary, these findings demonstrate a specific role for chronically upregulated Schwann cell-derived signaling of the NRG1-I isoform in precipitating key pathological hallmarks of peripheral neuropathies, including onion bulb structures.

**Glial NRG1-I signaling promotes survival and attraction of Schwann cells**. Which cellular mechanism underlies the formation of onion bulb structures downstream of the specific glial

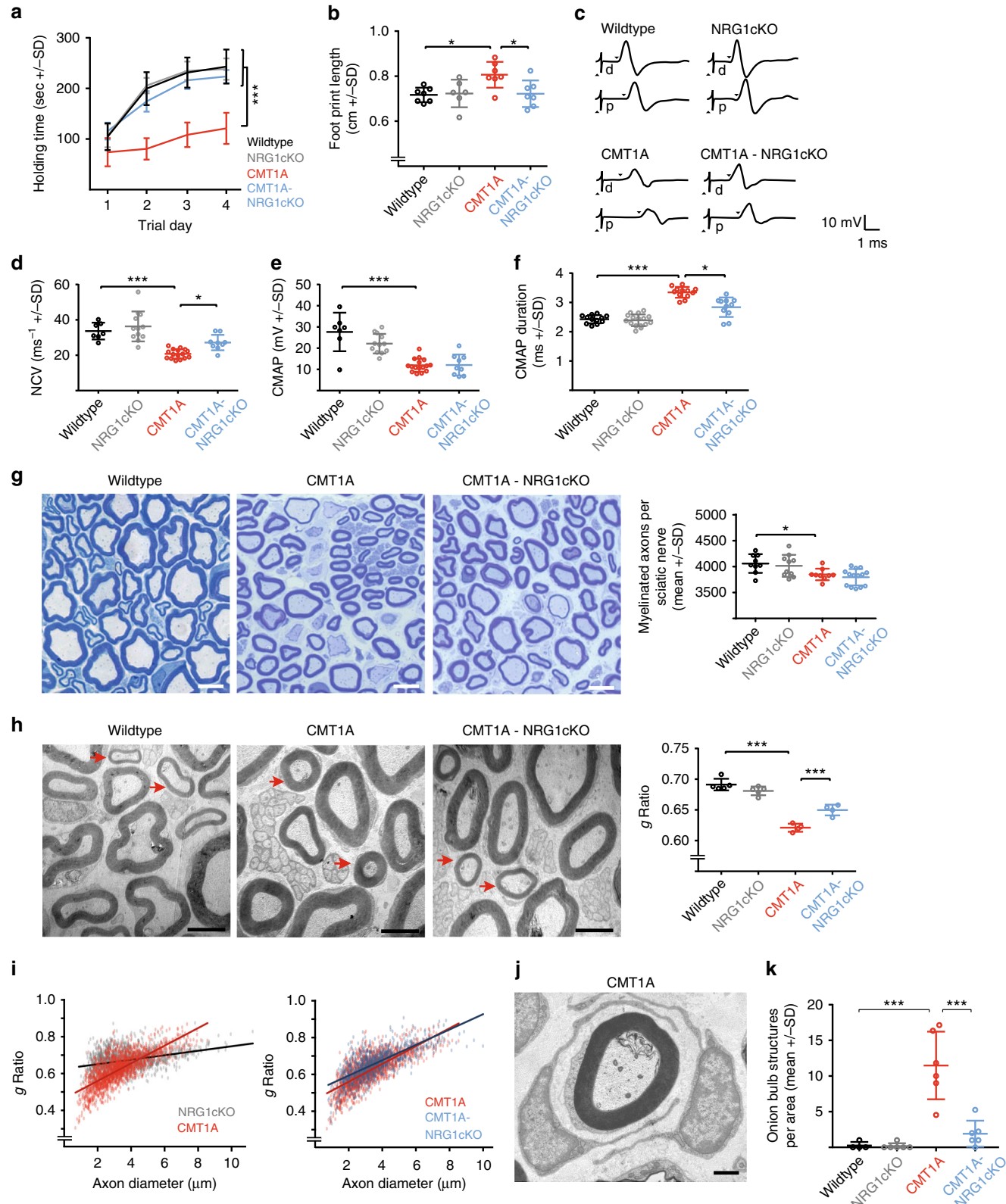

NRG1-I expression? To investigate the pathomechanism of onion bulb formations, we first addressed the hypothesis that onion bulb formations require the generation of supernumerary Schwann cells[13–15,36]. Of note, supernumerary Schwann cells in CMT1A are already present during early postnatal development[37] and persist throughout the disease course. Can supernumerary Schwann cells be explained by a decreased rate of apoptosis that normally eliminates excess Schwann cells in the first weeks of postnatal development, and is this a function of glial NRG1-I as a survival factor[38,39]?

When we cultured primary Schwann cells in growth factor-free medium (i.e., in the absence of axons), we found a decreased number of Schwann cells in cultures derived from NRG1cKO mutants, similar to control cultures treated with a

**Fig. 2** Glial Neuregulin-1 (NRG1) ablation improves neuropathic symptoms in adult CMT1A mice. **a** Rotarod analysis of 3-month-old mice on 4 consecutive days (wild type $n = 7$, NRG1cKO $n = 8$, CMT1A $n = 7$, CMT1A-NRG1cKO $n = 8$, two-way analysis of variance (ANOVA) with Tukey's multiple comparison tests). **b** The foot print length in 4-month-old CMT1A ($n = 7$), wild type ($n = 6$), NRG1cKO ($n = 6$), and CMT1A-NRG1cKO ($n = 7$) mice. **c** Representative electrophysiological traces of sciatic nerve recordings of 4-month-old wild-type, NRG1cKO, CMT1A and CMT1A-NRG1cKO mice after distal (d) and proximal (p) stimulation (up arrow: stimulus artefact, down arrow: distal motor latencies). **d** Quantification of nerve conduction velocity (NCV) in 4-month-old wild type ($n = 7$), NRG1cKO ($n = 12$), CMT1A ($n = 16$), and CMT1A-NRG1cKO ($n = 9$) mice. **e** Quantification of compound muscle action potential (CMAP) from mice analyzed in **d**. **f** Quantification of temporal dispersion as measured by CMAP duration in 4-month-old wild type ($n = 12$), NRG1cKO ($n = 16$), CMT1A ($n = 12$), and CMT1A-NRG1cKO ($n = 11$) mice. The average values from four stimulations (distal and proximal from both sides) were calculated per mouse. **g** Representative light microscopic images of sciatic nerve cross-sections from 4-month-old wild-type, CMT1A, and CMT1A-NRG1cKO adult mice (left panels, scale bar 5 µm). Quantification of the number of myelinated axons per sciatic nerve cross-section (right panel) in wild type ($n = 7$), CMT1A ($n = 6$), and CMT1A-NRG1cKO ($n = 7$) mice. **h** Representative electron micrographs of sciatic nerve cross-sections from 4-month-old mice (left panels), demonstrating hypermyelination (arrows) of small caliber axons in CMT1A compared to wild-type and CMT1A-NRG1cKO mutants (scale bar 2.5 µm). Quantification of the mean myelin sheath thickness (g-ratio) (right panel, wild type $n = 5$, NRG1cKO $n = 5$, CMT1A $n = 5$, CMT1A-NRG1cKO $n = 5$, 221–375 fibers were measured per animal). **i** Myelin sheath thickness (g-ratio, data of **f**) plotted against the axon diameter shows hypermyelination of small to mid-caliber and hypomyelination of large caliber fibers in CMT1A mice compared to controls (NRG1cKO, left panel). Ameliorated hypermyelination of small to mid-caliber axons is visible in CMT1A-NRG1cKO mice when compared to CMT1A mice (right panel). **j** Electron micrograph of an onion bulb structure in sciatic nerves of a 4-month-old CMT1A mouse (scale bar 1 µm). **k** Quantification (right panel) of onion bulb structures per area (15,740 µm²) in cross-sections of sciatic nerve of 4-month-old mice (wild type $n = 4$, NRG1cKO $n = 6$, CMT1A $n = 6$, CMT1A-NRG1cKO $n = 6$). Source data are provided as a source data file. All respective p-values are depicted as a range of significance with *$p < 0.05$ and ***$p < 0.001$; **b**, **d**, **e**–**h**, **k**: one-way ANOVA, Tukey's post test

pharmacological ErbB2 receptor inhibitor (Fig. 4a, b). In line, the number of apoptotic Schwann cells as assessed by terminal deoxinucleotidyl transferase-mediated dUTP-fluorescein nick end labeling (TUNEL) assay was increased in NRG1cKO cultures (Fig. 4c), suggesting that glial NRG1 promotes Schwann cell survival. In support of this notion, we found no evidence for NRG1-mediated increased Schwann cell proliferation in CMT1A mice (Supplementary Fig. 3a) but confirmed reduced Schwann cell apoptosis in CMT1A mice in vivo, whereas Schwann cell apoptosis was strongly increased in CMT1A-NRG1cKO mutants (Fig. 4d). As a result, Schwann cell number was normalized in CMT1A-NRG1cKO mice (Fig. 4e). Finally, we observed an increased Schwann cell number and a concomitant reduction of TUNEL-positive Schwann cells in sciatic nerves of NRG1-I OE mice (Fig. 4d, f).

NRG1 serves as a chemoattractant[40] and supports Schwann cell motility[23,41] and therefore could mediate the concentric alignment of supernumerary Schwann cells around the axon-associated innermost Schwann cell within onion bulb formations. A Boyden chamber migration assay comparing the migration of wild-type Schwann cells towards Schwann cells derived from either wild-type or NRG1cKO mice revealed that glial NRG1 acts as a paracrine attraction signal for Schwann cells (Fig. 4g). In addition, isoform-specific BaseScope in situ hybridization for *Nrg1-I* in young CMT1A mice identified *Nrg1-I* transcription predominantly in the axon-associated, myelinating Schwann cell, in line with our immunohistochemical findings (Figs. 1f and 4h and Supplementary Fig. 3b, c). In line, supernumerary Schwann cells were found to be closely associated with *Nrg1-I*-over-expressing myelinating Schwann cells in NRG1-I OE mice in vivo (Fig. 4i). Taken together, these findings suggest that NRG1-I serves as a paracrine signal expressed by the innermost Schwann cell that initially promotes Schwann cell survival and subsequently attracts and aligns supernumerary Schwann cells during the formation of onion bulb structures.

**Schwann cell-derived NRG1-I induces glial MEK/ERK signaling.** To investigate molecular signaling pathways downstream of glial NRG1-I overexpression in CMT1A, we employed Phospho Explorer antibody arrays to compare peripheral nerve lysates from control, CMT1A, and CMT1A-NRG1cKO mice. This unbiased approach identified NRG1/ErbB and mitogen-activated protein kinase (MEK/ERK) signaling among the most

prominently regulated pathways (Fig. 5a). We biochemically confirmed ErbB2 receptor upregulation and phosphorylation as well as MEK/ERK hyperactivity in adult sciatic nerves from CMT1A mice (as previously reported[16]) and found both conditions to be normalized in CMT1A-NRG1cKO mice (Fig. 5b). Accordingly, ErbB2 and MEK/ERK signaling activity were increased in NRG1-I OE mice (Fig. 5c), consistent with a direct role of glial NRG1-I in the stimulation of MEK/ERK signaling, independent from PMP22-mediated CMT1A pathology. A paracrine and autocrine signaling axis comprising glial NRG1-I/ErbB2/MEK/ERK was further supported by decreased MEK/ERK activity in primary Schwann cell cultures from CMT1A mice in response to pharmacological inhibition of ErbB2 receptor activity (Fig. 5d).

Glial MEK/ERK signaling constitutes an important regulator of Schwann cell dedifferentiation[19] and mediates Schwann cell dysdifferentiation in CMT1A[16,42]. Notably, the ablation of Schwann cell-derived *Nrg1* in CMT1A-NRG1cKO mice normalized the expression of immaturity- and dedifferentiation-associated markers (such as *cJun* and *Sox2*). However, the expression of the genuine dedifferentiation markers *Olig1* and *Shh*[18] was not reduced (Fig. 5e), indicating that only partial dedifferentiation is a consequence of glial NRG1-I expression in CMT1A. As both supernumerary and myelinating Schwann cells in CMT1A express immaturity markers, their reduced expression in CMT1A-NRG1cKO mice may result from a lower number of supernumerary (immature) Schwann cells (Fig. 4e) and a decreased aberrant expression of the respective genes in individual myelinating cells. In line with the latter, an immunohistochemical analysis revealed that the number of cJUN-positive nuclei of myelinating Schwann cells was reduced in CMT1A-NRG1cKO mice (Fig. 5f). Accordingly, cJUN was expressed by supernumerary and by mature myelinating Schwann cells in NRG1-I OE mice (Fig. 5f).

Notably, peripheral nerves from NRG1-I OE mice displayed an increased expression of Schwann cell immaturity markers but not a fully dedifferentiated Schwann cell state (as determined by unaltered *Olig1* and *Shh* expression[18]) (Fig. 5e), consistent with the hypothesis that hyperstimulated NRG1-I signaling alone is not sufficient to induce a fully dedifferentiated state in Schwann cells. We previously suggested that Schwann cell dedifferentiation in CMT1A results from dysbalanced pathway activity, with increased MEK/ERK and decreased phosphoinositide-3 kinase (PI3K)/AKT signaling[16]. In line with incomplete dedifferentiation, NRG1-I OE

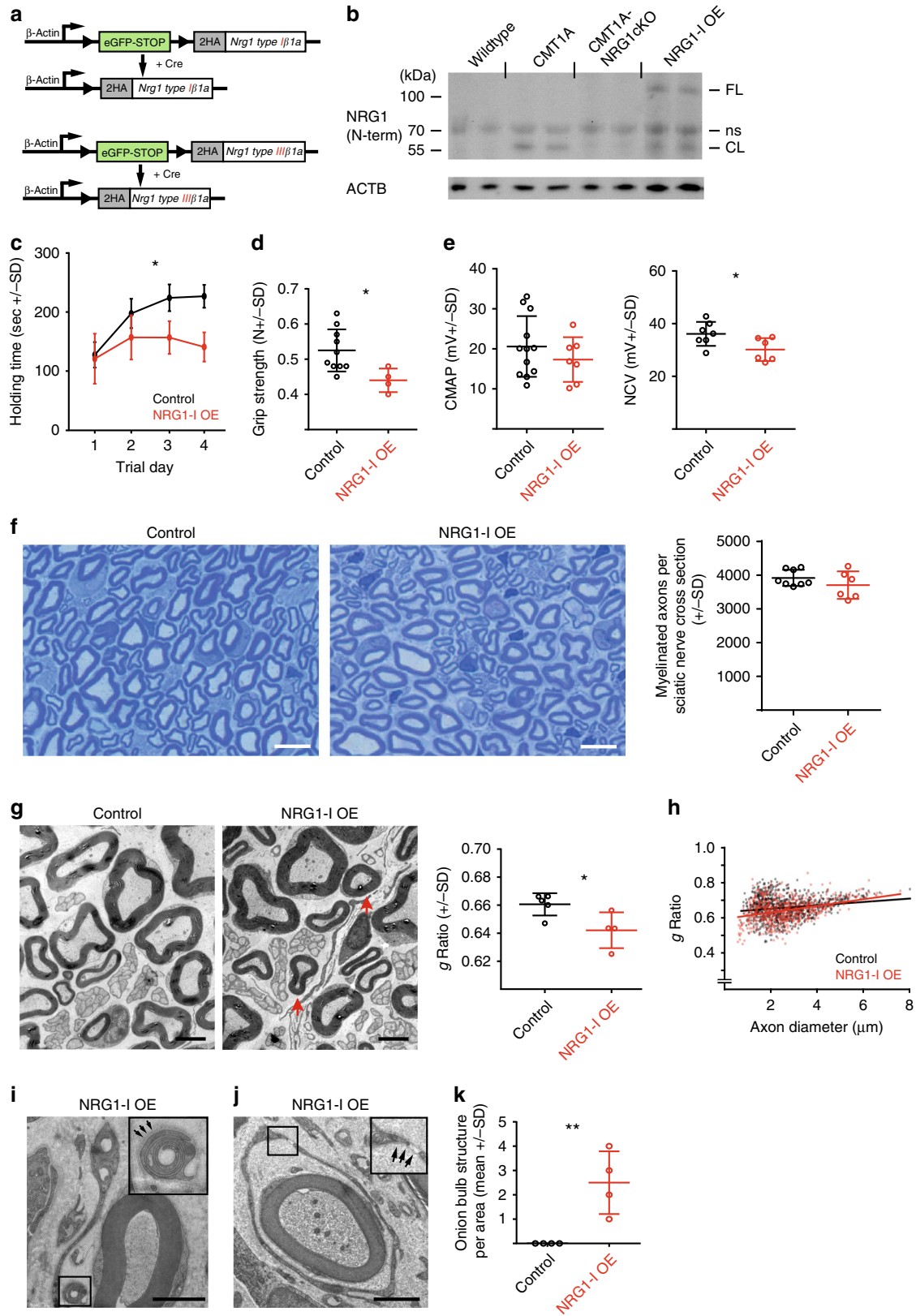

mice were characterized by a concomitant activation of both MEK/ERK and PI3K/AKT signaling (Fig. 5c, Supplementary Fig. 3d). Moreover, AKT phosphorylation (in contrast to MEK/ERK) was not normalized in CMT1A-NRG1cKO mice (Fig. 5b, Supplementary Fig. 3e), consistent with an independent defect of

*PMP22* transgenic Schwann cells to upregulate PI3K/AKT signaling[16].

**Dedifferentiation triggers onion bulbs downstream of glial NRG1-I.** Could the observed differences in the differentiation

**Fig. 3** Neuregulin-1 type 1 (NRG1-I) overexpression (OE) in Schwann cells triggers neuropathic symptoms. **a** Genetic constructs for mice overexpressing *Nrg1-Iß1* (upper panel) and *Nrg1-IIIß1* (lower panel) isoforms after *Cre*-mediated recombination, respectively. The constructs are composed of an eGFP-stop cassette, flanked by loxP sites following the murine *Nrg1-Iß1* or *Nrg1-IIIß1* cDNA fused to an HA tag, respectively. The transgene expression is driven by a beta actin promoter. **b** Western blot of sciatic nerve lysates from 4-month-old wild-type, CMT1A, CMT1A-NRG1cKO, and NRG1-I OE mice with N-terminal NRG1 antibody (SC393006) shows expression of the cleaved (CL) N-terminal fragment of NRG1 protein in CMT1A mice and in NRG1-I OE mice (next to full-length [FL] NRG1, ns: not Schwann cell specific; for full blots, see Supplementary Figure 5). **c** Rotarod analysis of 3-month-old mice on 4 consecutive days (control $n = 10$, NRG1-I OE $n = 4$, two-way analysis of variance). **d** Hind limb grip strength test of 3-month-old mice (control $n = 10$, NRG1-I OE $n = 4$; Student's $T$ test). **e** Nerve conduction velocity (NCV, right panel) and compound muscle action potential (CMAP, left panel) in 4-month-old mice (CMAP: control $n = 12$, NRG1-I OE $n = 7$, NCV: control $n = 7$, NRG1-I OE $n = 6$, Student's $T$ test). **f** Representative light microscopic images of sciatic nerve cross-sections of 4-month-old NRG1-I OE and respective control mice (left panels, scale bar: 10 μm). Quantification of the number of myelinated axons per sciatic nerve cross-section (right panel, control $n = 8$, NRG1-I OE $n = 6$, Student's $T$ test). **g** Representative electron micrographs of sciatic nerve cross-sections of 4-month-old NRG1-I OE (middle panel) and control mice (left panel, scale bars 2.5 μm), demonstrating hypermyelination of small to mid-caliber axons (arrows). Quantification of g-ratio (right panel, control $n = 5$, NRG1-I OE $n = 4$, Student's $T$ test). **h** Scatter blots of quantified myelin sheath thickness (g-ratio, data from **g**) plotted against the axon diameter. **i** Electron micrograph of aberrant Schwann cell protrusions in a femoral nerve of an 11-month-old NRG1-I OE mouse (scale bar 2 μm). **j** Electron micrograph of an onion bulb-like structure (right) from **i** (scale bar 2 μm). Arrows in blow up in **i** and **j** demarcate the basal lamina, which identifies a Schwann cell as such. **k** Quantification of onion bulb structures per cross section area (15,740 μm²) in sciatic nerves of 4-month-old NRG1-I OE ($n = 4$) mice compared to controls ($n = 4$; Student's $T$ test). Source data are provided as a source data file. All respective $p$ values are depicted as a range of significance with *$p < 0.05$; **$p < 0.01$

state of Schwann cells in CMT1A and NRG1-I OE mice explain why onion bulbs were more frequent in CMT1A mice compared to the occasional onion bulb formations in NRG1-I-overexpressing mice? We hypothesized that Schwann cell dedifferentiation may serve as an additional trigger for the formation of onion bulb structures. Consequently, in CMT1A, the more complete dedifferentiation status of Schwann cells would promote their responsiveness to NRG1-I signaling and the subsequent formation of onion bulb structures. Nerve injury causes pronounced Schwann cell dedifferentiation[18], and repetitive nerve injury has been suggested to induce onion bulb structures[36]. We hence postulated that nerve injury in NRG1-I OE mice may serve as a final trigger to fully dedifferentiate Schwann cells, such that supernumerary Schwann cells show enhanced responsiveness to NRG1 signaling.

Remarkably, a single nerve crush was sufficient to generate numerous onion bulb formations in NRG1-I OE mice but not in wild-type controls (Fig. 6a). When we performed the same experiment in CMT1A mice, characterized by endogenous *Nrg1-I* upregulation, we were likewise able to boost the generation of onion bulb formations when compared to injured wild-type controls (Fig. 6b). As predicted, this effect was lost in CMT1A mice with ablated glial *Nrg1* (Fig. 6b), further demonstrating the essential role of Schwann cell-derived NRG1-I for the generation of onion bulb structures.

**Limited axon–glia interactions contribute to glial NRG1-I expression**. Finally, we asked what drives aberrant NRG1-I expression in CMT1A Schwann cells? Axonal NRG1 signaling controls physiological glial MEK/ERK activity during postnatal myelination[43–45], and we showed previously that loss of axonal NRG1-mediated MEK/ERK signaling in Schwann cells after peripheral nerve crush induces glial NRG1-I expression[29]. In line, also in primary rat Schwann cells, the pharmacological inhibition of MEK/ERK induces *Nrg1-I* mRNA expression (Fig. 6c). Indeed, we observed a pronounced decrease in MEK/ERK activity early postnatally in CMT rats[16], which precedes the upregulation of Schwann cell *Nrg1* mRNA (Fig. 1d). Hence, we speculated that increased NRG1-I expression in CMT1A Schwann cells may result from an impaired access to or integration of physiological levels of axonal NRG1 signaling early postnatally during the onset of myelination.

To test this concept, we genetically increased axonal NRG1 levels in CMT1A mice by breeding transgenic mice with neuronal overexpression of NRG1. Resulting double transgenic mice

indeed showed a reduction of glial *Nrg1-I* expression and a decreased number of onion bulb formations compared to CMT1A controls (Fig. 6d, e). In line with the dependence of NRG1-I induction from myelination, we found that the upregulation of NRG1-I in CMT1A myelinating co-cultures followed the time course of myelination and correlated with impaired longitudinal myelin growth in vitro (Fig. 6f). Notably, we observed a dramatically reduced internodal length in vivo in CMT1A animals from early postnatal development on (Fig. 6g, Supplementary Fig. 2a). Consequently, *Pmp22* transgenic Schwann cells may suffer from a decreased contact area with the corresponding axon that is likely to expose individual Schwann cells to less axonally presented NRG1 during postnatal development.

In line, we have previously shown that a reduction of axonal NRG1 type III on the axonal surface in *Nrg1-III* heterozygous animals is sufficient to induce glial NRG1-I expression during postnatal myelination in otherwise healthy animals[29]. When we analyzed *Nrg1-III* heterozygous animals at the age of 11 months, however, we no longer found glial *Nrg1-I* mRNA expression to be increased (Fig. 6h), suggesting that additional factors are necessary to maintain a long-term glial *Nrg1-I* expression. In favor of this concept, we detected a further increase in glial *Nrg1-I* expression in dedifferentiated Schwann cells in CMT1A mice crossbred to *Nrg1-III* heterozygous animals at progressive disease stages (11 months of age, Fig. 6h). Moreover, *Nrg1-III* heterozygous CMT1A double mutants displayed a significantly higher number of onion bulbs compared to single CMT1A mutants at 11 months of age (Fig. 6i).

Hence, axonal NRG1 levels still contribute to glial *Nrg1-I* induction at later disease stages in CMT1A. We therefore hypothesize that demyelination and remyelination events in late CMT1A disease with dedifferentiated Schwann cells remyelinating mature, preexisting axons perpetuates the ontogenetic importance of the NRG1 axon–glia interaction. Indeed, mature axons are characterized by a limited expression of NRG1 on the axonal surface[28]. As a consequence, Schwann cell-derived NRG1-I expression should parallel disease onset in other demyelinating neuropathies. Indeed, a longitudinal analysis of the slowly progressive, late-onset demyelinating CMT1B mouse model, caused by a heterozygous deletion of the *Mpz* gene[46,47], revealed an upregulation of *Nrg1-I* only during late disease stages (after P180), when demyelinating and remyelinating events become prominent (Supplementary Fig. 3f). In line, primary myelination is not defective and internodal length is unaltered in CMT1B mice[46,47] (Supplementary Fig. 2a). In contrast, mice homozygous

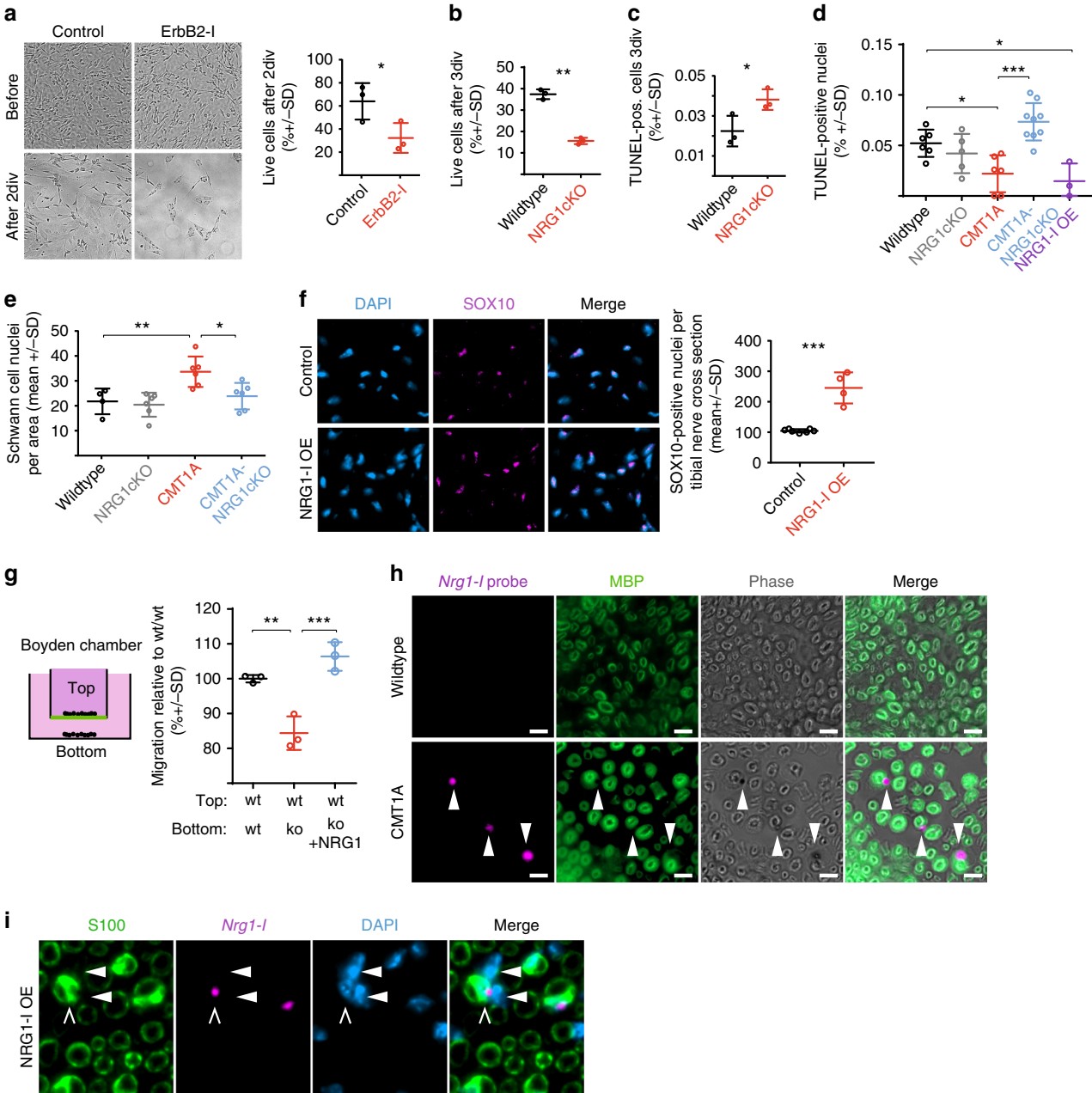

**Fig. 4** Schwann cell Nrg1 promotes survival and migration of Schwann cells. **a** Survival assay with primary rat Schwann cells after 2 days in vitro (2div) with or without an ErbB2 inhibitor (CAS 928207-02-7, 30 μM). Representative before/after pictures (left panels) with quantification (right panel; **a–c**: $n = 3$ independent Schwann cell preparations per group, paired Student's T test). **b** Survival assay with primary mouse Schwann cells from wild-type and NRG1cKO mice after 3div. **c** In the same experiment as in **b**, the apoptosis rate was assessed by terminal deoxinucleotidyl transferase-mediated dUTP-fluorescein nick end labeling (TUNEL) assay. **d** Quantification of Schwann cell apoptosis as measured by percentage of TUNEL-positive nuclei on sciatic nerve longitudinal sections from 11-day-old mice (wild type $n = 6$, NRG1cKO $n = 5$, CMT1A $n = 6$, CMT1A-NRG1cKO $n = 9$, NRG1-I OE $n = 3$, one-way analysis of variance (ANOVA), Tukey's post test). **e** Quantification of Schwann cell nuclei per area (15,740 μm²) on electron micrographs of 4-month-old wild type ($n = 4$), NRG1cKO ($n = 6$), CMT1A ($n = 6$), and CMT1A-NRG1cKO ($n = 6$) mice (one-way ANOVA and Tukey's post test). **f** Fluorescent immunohistochemistry on tibial nerve cross-sections from 4-month-old control ($n = 4$) and Neuregulin-1 type 1 (NRG1-I) overexpression (OE) ($n = 3$) mice against SOX10 (left panels) and respective quantification (right panel, Student's T test). **g** Schematic set-up of Boyden chamber migration experiments (left panel) and respective quantification (right panel) of the percentage of cells that migrated from the top compartment through the mesh (green in left panel). The top compartment was always seeded with wild-type (wt) Schwann cells, whereas in the bottom compartment either with wild type (wt, black, set to 100%) or NRG1cKO Schwann cells were seeded and the latter without (ko, red) or with (ko+ NRG1, blue) recombinant NRG1 (quantification of three independent experiments, one-way ANOVA, Tukey's post test). **h** BaseScope[TM] in situ hybridization with a *Nrg1-I* probe (magenta) on sciatic nerve cross-sections of 18-day-old wild-type and CMT1A mice. Myelin was counterstained with MBP (green) and together with the phase contrast demonstrates *Nrg1-I* expression in myelinating Schwann cells in the CMT1A mouse (arrows, scale bar 5 μm). **i** BaseScope in situ hybridization with a *Nrg1-I* probe (magenta) on sciatic nerve cross-sections of a 4-month-old NRG1-OE mouse showing expression of *Nrg1-I* (magenta) in Schwann cells (S100, green). Note that supernumerary Schwann cells (closed arrows) are located nearby a *Nrg1-I*-expressing myelinating Schwann cell (open arrow, scale bar 5 μm, DAPI: nuclei). Source data are provided as a source data file. All respective p values are depicted as a range of significance with *$p < 0.05$; **$p < 0.01$; and ***$p < 0.001$

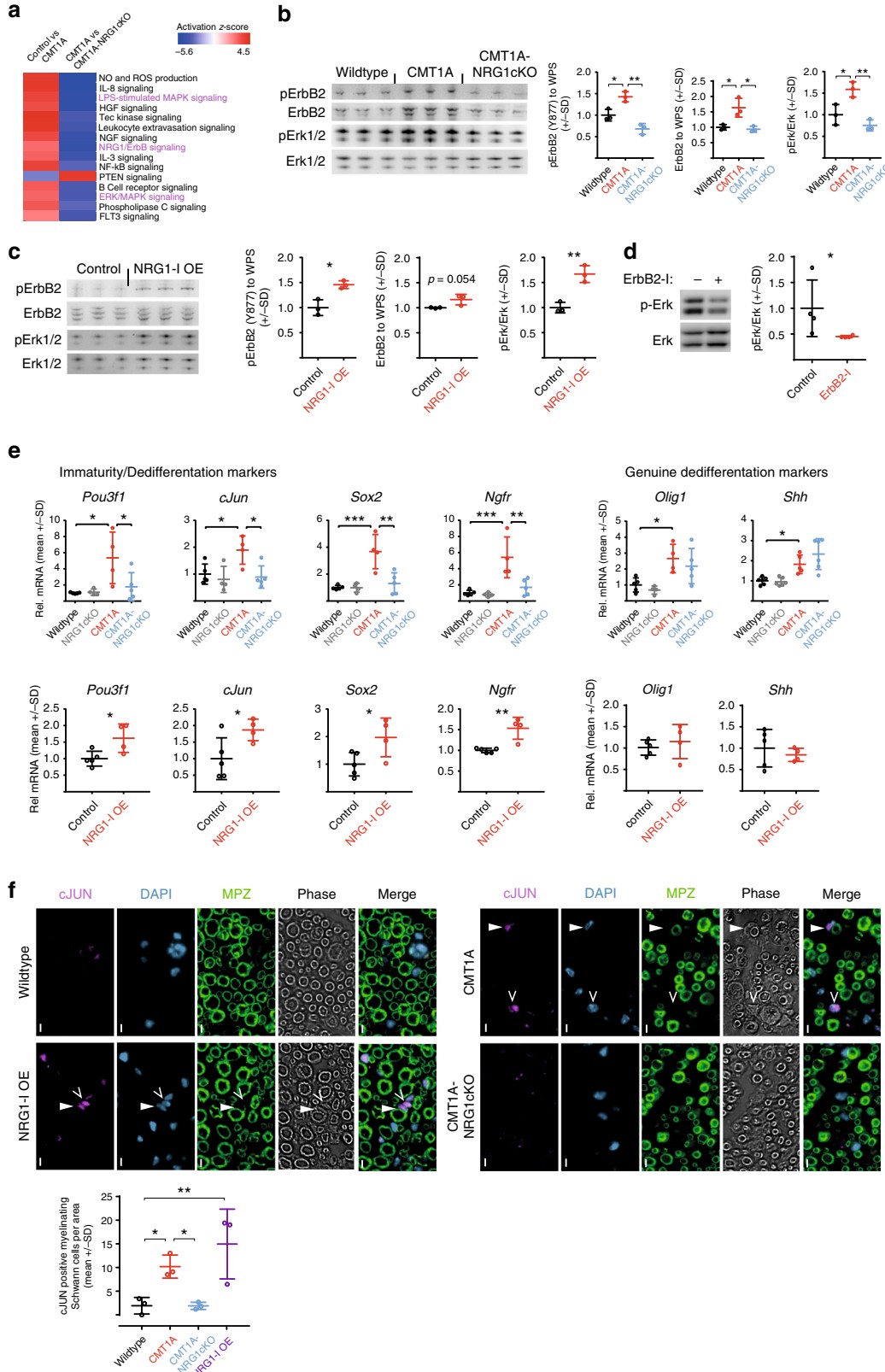

for a null mutation in the same gene (*Mpz*) upregulate *Nrg1-I* mRNA at earlier disease stages, in accordance with an early-onset severe hypomyelination (Fig. 1a). Together, these data suggest variable timing of distinct disease-associated processes that act upstream of NRG1-I induction.

## Discussion

Our findings identify the induction of the NRG1-I isoform in Schwann cells as an integral part of a final common glial response in CMT1A and most likely various other demyelinating polyneuropathies. In case of CMT1A, we demonstrate that auto-

**Fig. 5** Schwann cell Neuregulin-1 type 1 (NRG1-I) activates ErbB2 and Mek/Erk signaling. **a** Phospho protein explorer antibody array with sciatic nerve protein extracts from 30-day-old mice. Heat map of the top 15 pathways that show inverse regulation between comparison one (NRG1cKO versus CMT1A, left lane) and two (CMT1A versus CMT1A-NRG1cKO, right lane), ranked by activation Z-score. Nrg1/ErbB2 and Erk/MAPK signaling (highlighted in magenta) were found among the top deregulated pathways, two technical replicates of $n = 5$ pooled samples per genotype were performed and data were analyzed by Ingenuity Pathway Analysis (p value cutoff: log 45, see Methods for details). **b** Western blot analyses (left panels) and respective quantifications (right panels) of sciatic nerve lysates of 4-month-old mice showing increased expression and phosphorylation of ErbB2 (normalized to whole-protein-stained membrane, see Supplementary Fig. 5) and increased phosphorylation of Erk1/2 in CMT1A compared to wild-type mice. Both ErbB2 and Erk1/2 phosphorylation are normalized in CMT1A-NRG1cKO mice ($n = 3$ per group, one-way analysis of variance (ANOVA) and Tukey's post test; for full blots, see Supplementary Fig. 5). **c** Western blot analyses (left panels) and respective quantifications (right panels) of sciatic nerve lysates of 4-month-old mice showing hyperphosphorylation of ErbB2 (normalized to whole-protein-stained membrane, see Supplementary Figure 5) and Erk1/2 in NRG1-I OE mice compared to controls ($n = 3$ per group, Student's T test). Western blots of ErbB2 and pErbB2 were conducted in parallel with the same samples from the same protein preparations on two different membranes. **d** Western blot of primary CMT1A rat Schwann cells against phosphorylated and constitutive Erk1/2. Schwann cells were treated with or without ErbB2 inhibitor (10 μM, CAS 928207-02-7) for 6 h, which resulted in decreased phosphorylation of Erk1/2. Quantification of four independent experiments is shown (right panel, paired Student's T test). **e** Relative mRNA expression of dedifferentiation and immaturity markers in sciatic nerve lysates (upper row: wild type $n = 5$, NRG1cKO $n = 4$, CMT1A $n = 4$, CMT1A-NRG1cKO $n = 5$, one-way ANOVA with Tukey's post test; lower row: control $n = 5$, NRG1-I OE $n = 4$, Student's T test). **f** Immunohistochemical analysis of the cJUN in sciatic nerve cross-sections from 4-month-old wild-type, CMT1A, CMT1A-NRG1cKO, and NRG1-I OE mice. Representative pictures are shown in the upper panels displaying cJUN expression (magenta, myelin MPZ in green) in myelinating (closed arrows) and supernumerary (open arrows) Schwann cells in CMT1A and NRG1-OE mice. Quantification of cJUN-positive nuclei from myelinating Schwann cells is shown in the panel below (control, $n = 4$; CMT1A, $n = 3$; CMT1A-NRG1cKO mice, $n = 3$; NRG1-I OE, $n = 3$; one-way ANOVA with Tukey's post test). Source data are provided as a source data file. All respective p values are depicted as a range of significance with *$p < 0.05$; **$p < 0.01$; and ***$p < 0.001$

paracrine NRG1-I signaling in Schwann cells is the underlying molecular pathomechanism for hypermyelination and onion bulb pathology. Onion bulbs are a hitherto unexplained key hallmark of Schwann cell pathology in demyelinating peripheral neuropathies[10–13]. Using Schwann cell-specific NRG1 loss- and gain-of-function mouse mutants, we show that onion bulbs result from a persistent glial NRG1-I expression by the most inner Schwann cell that contacts the associated axon. While we found NRG1-I to be predominantly expressed by the innermost Schwann cell at early disease stages in CMT1A, an additional expression by supernumerary Schwann cells at progressive disease stages, when axonal loss and demyelination and remyelination become apparent, is likely.

While supernumerary Schwann cells in wild-type mice are normally eliminated within the first postnatal weeks, they persist in peripheral nerves of CMT1A mutants due to glial NRG1-I-mediated survival functions. Subsequently, fully dedifferentiated (not simply immature) supernumerary Schwann cells[18] respond to paracrine glial NRG1-I attraction signals and concentrically align in onion bulb structures. Fully dedifferentiated Schwann cells are present in CMT1A disease from early development on, due to a dysbalanced activity of PI3K/AKT and MEK/ERK signaling[16]. However, Schwann cell dedifferentiation is also the result of acute nerve injury[18] and demyelination and remyelination cycles in adult peripheral nerves, in line with our finding that acute peripheral nerve crush strongly boosted the formation of onion bulb structures in NRG1-OE mice. In conclusion, persistent glial NRG1-I signaling originating from the innermost Schwann cells in combination with Schwann cell dedifferentiation in diseased peripheral nerves promotes the formation of onion bulb structures in CMT1A.

Why is glial NRG1-I induced in CMT1A disease and various demyelinating neuropathies? We previously demonstrated that glial NRG1-I is transiently upregulated after acute nerve injury, when Schwann cells encounter a de facto loss of axonal contact[29]. Here we propose a model according to which *Pmp22* over-expression compromises axo-glial NRG1 signaling in early CMT1A disease that results in functionally denervated Schwann cells that respond with aberrant glial *NRG1-I* induction (see model, Supplementary Fig. 4). This concept is supported by dramatically reduced internodal length in CMT1A disease from early postnatal development on, and a concomitant glial NRG1-I

induction that parallels the onset of myelination in vivo and in vitro. Furthermore, we demonstrate that CMT1A Schwann cells dynamically adapt NRG1-I expression in response to altered axonal NRG1 levels in both axonal *Nrg1* overexpressing and haploinsufficient mutant mice. However, we argue that changes in axonal NRG1 levels alone, i.e., without a concomitant pathological trigger, are not sufficient to drive a persistent glial NRG1-I expression in adult nerves. Indeed, we suggest that repetitive cycles of demyelination and remyelination that are characteristic of progressive CMT1A disease[11–13] account for glial NRG1-I induction in adult diseased nerves. Here dedifferentiated Schwann cells remyelinate mature axons, which are characterized by a limited expression of NRG1-III[27,28], in contrast to growing axons during development or after mechanical injury. Hence, in contrast to acute nerve injury, where a transient upregulation of Schwann cell-derived *Nrg1-I* is subsequently inhibited by axonal regrowth and concomitant re-expression of axonal NRG1[29], this does not occur in chronic demyelinating neuropathies.

In line, we demonstrate that *Nrg1-I* upregulation correlates with demyelination and remyelination events during late disease stages (P365) in CMT1B mice, which show no defects in primary myelination as compared to CMT1A disease. Thus the heterogeneity in both onset and dynamics of glial NRG1-I expression could explain the diversity of molecular Schwann cell responses and nerve histopathology in the context of different demyelinating neuropathies.

Genetic disruption of glial NRG1 in CMT1A-NRG1cKO mice improved not only the histopathological disease hallmarks but also the disease phenotype of adult CMT1A animals.

Notably, the pronounced amelioration of the clinical phenotype in double mutants is accompanied by only a partial improvement on the electrophysiological level, likely reflecting a non-linear correlation between the clinical phenotype and electrophysiology, an observation also described for human patients[7,48,49]. We also note that the electrophysiological recordings characterize the entire sciatic nerve, whereas the clinical phenotype depends on the functional integrity of specific axonal subsets. Nevertheless, we observed a significantly improved NCV in 4- and 11-month-old CMT1A-NRG1cKO mice, which is normally driven by large caliber myelinated axons. Indeed, the NCV slowing characteristic of human CMT1A nerves and respective animal models has been attributed to the

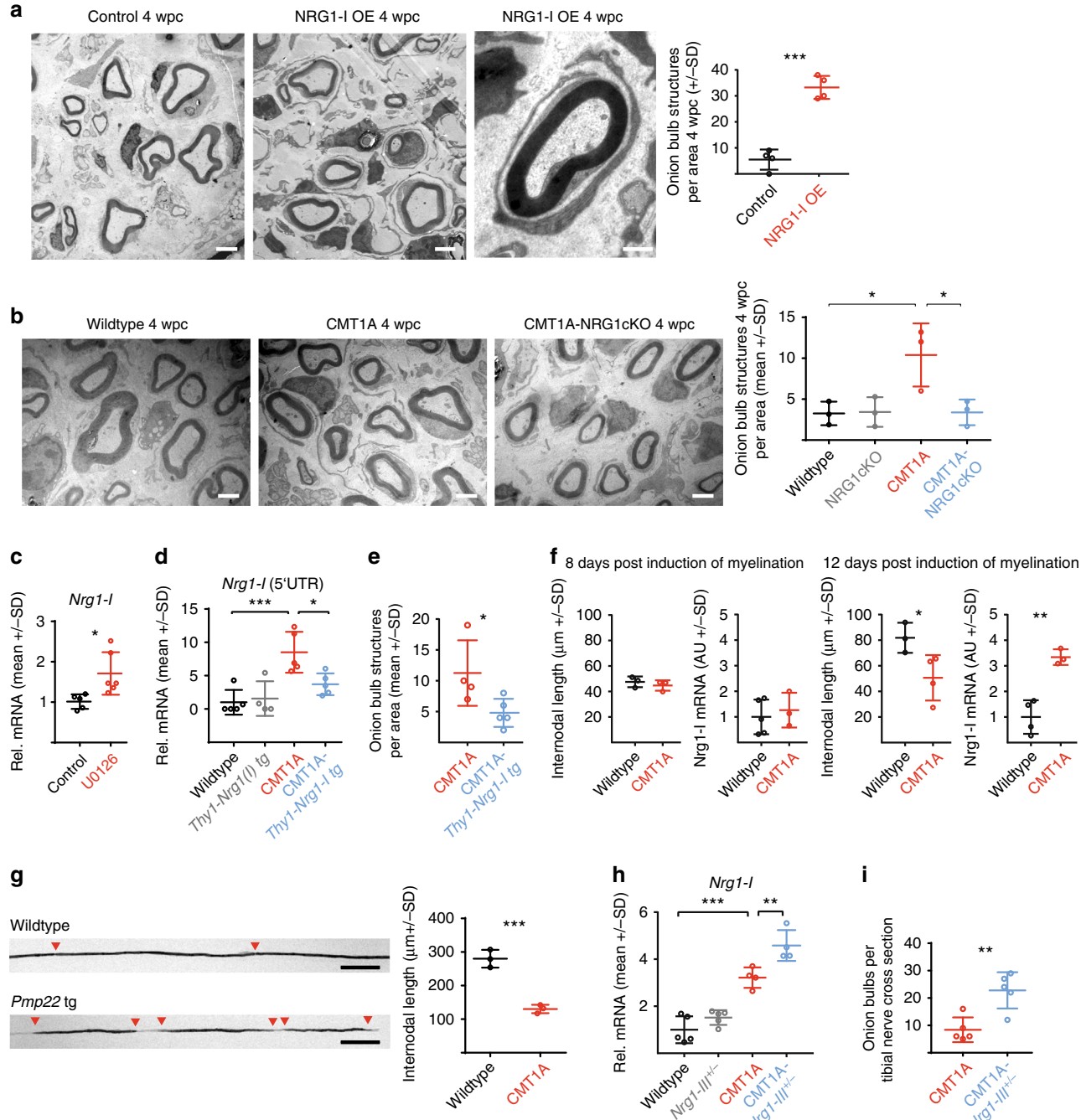

hypomyelination of large fibers in CMT1A disease[6]. However, the basal characteristics of myelinated fibers in CMT1A nerves are fundamentally different from wild type, which is likely to impact on the fiber types measured in electrophysiological recordings. Indeed, the overall axonal diameter is reduced in CMT1A and large caliber axons suffer from demyelination, suggesting that smaller caliber axons are more likely to contribute to electro-physiological measurements in CMT1A. Hence, the improved hypermyelination of small to mid-caliber axons in CMT1A-NRG1cKO mice may, at least partially, contribute to the improved nerve conduction in double mutants. Next to the changes in NCV, we note an improved temporal dispersion but no CMAP change in 4-month-old CMT1A-NRG1cKO animals, while at later disease stages, the CMAP is also ameliorated in double mutants. As the numbers of myelinated axons as such are not improved upon glial NRG1-I ablation, a complex interplay

between the histopathological disease hallmarks and axonal function in CMT1A is probable. Indeed, the improved axonal diameter distribution in CMT1A-NRG1cKO animals that becomes present only later in disease (i.e., at 11 months of age) may provide a potential mechanism for functional impairment in CMT1A disease. However, in a progressive disease it becomes increasingly difficult to distinguish cause and consequences of individual pathological parameters.

We previously demonstrated that the treatment of CMT1A rats with recombinant NRG1, when restricted to early postnatal development, improves the CMT1A disease phenotype[16], in line with our present findings on the stimulatory role of Schwann cell-derived NRG1-I in CMT1A, during early postnatal stages. Thus exogenous as well as Schwann cell-derived NRG1 may adopt axonal NRG1 type III function in a timely restricted manner during early postnatal development and promote peripheral

**Fig. 6** Neuregulin-1 type 1 (NRG1-I) induced interaction between Schwann cells mediates onion bulb formation. **a** Representative electron micrographs of sciatic nerve cross-sections 4 weeks post crush (4wpc) in a mouse overexpressing glial NRG1-I (middle panel with blow up of an onion bulb structure) and respective control (left panel, scale bars 2.5 µm, blow up 1 µm). Quantification per area (15,740 µm$^2$) is shown in the right panel ($n = 4$ per group, Student's $T$ test). **b** Representative electron micrographs of sciatic nerve cross-sections of 4-month-old wild-type, CMT1A, and CMT1A-NRG1cKO mice 4 weeks post crush (4wpc, scale bars 2.5 µm) and respective quantification of onion bulb-like structures per area (15,740 µm$^2$, $n = 3$ per group, one-way analysis of variance (ANOVA) and Tukey's post test). **c** Relative mRNA expression of *Nrg1-I* in primary rat Schwann cells is increased upon treatment for 12 h with the MEK inhibitor U0126 (10 µM) (red) compared to control (black) ($n = 5$ per group, Student's $T$ test). **d** Relative mRNA expression of *Nrg1-I* in sciatic nerves of 6-month-old mice (wild type, $n = 5$; *Thy1-Nrg1-I* tg, $n = 4$; CMT1A, $n = 5$; CMT1A-*Thy1-Nrg1-I*, $n = 5$). The quantitative PCR assay was designed to amplify the *Nrg1-I*-specific exon including a fragment of the 5'-untranslated region, which is not present in the *Thy1-Nrg1-I* transgene (one-way ANOVA and Tukey's post test). **e** Electron microscopic quantification of onion bulbs per sciatic nerve cross-section area (15,740 µm$^2$) in 6-month-old CMT1A ($n = 4$) and CMT1A-*Thy1-Nrg1-I* mice ($n = 5$, Student's $T$ test). **f** Internodal length and relative *Nrg1-I* mRNA expression in myelinating DRG neuron Schwann cell co-cultures 8 days (left panels) and 12 days (right panels) after induction of myelination (internodal length: wild type $n = 3$ and 3, CMT1A $n = 3$ and 4; *Nrg1-I* expression: wild type $n = 5$ and 4, CMT1A $n = 3$ and 3). The individual cultures were derived from separate embryos and are biological replicates, Student's $T$ test). **g** Representative images (left panels) of teased fibers from sciatic nerves of P18 old wild-type and CMT1A rats and quantification of internodal length (right panel, $n = 3$ per group, Students $T$ test, red triangles depict borders of myelin segments, scale bar 50 µm). **h** *Nrg1-I* mRNA expression in sciatic nerves of 11-month-old wild-type ($n = 5$) *Nrg1-III*$^{+/−}$ ($n = 5$), CMT1A ($n = 4$), and CMT1A-*Nrg1-III*$^{+/−}$ ($n = 5$) mice (one-way ANOVA with Tukey's post test). **i** Electron microscopic quantification of onion bulbs per tibial nerve cross-sections at the age of 11 months in CMT1A and CMT1A-*Nrg1-III*$^{+/−}$ mice ($n = 5$ per group, one-way ANOVA with Tukey's post test). Source data are provided as a source data file. All respective $p$ values are depicted as a range of significance with $^*p < 0.05$; $^{**}p < 0.01$; and $^{***}p < 0.001$

nerve myelination. Notably, the same Schwann cell response is beneficial after acute nerve injury, when survival of newly generated Schwann cells and their alignment referred to as the bands of Büngner promote nerve regeneration. However, in this scenario the regrowth of new axons subsequently terminates glial NRG1-I expression and prevents the persistent overstimulation of Schwann cells seen in CMT1A disease (see model, Supplementary Fig. 4). Here chronically active autoparacrine NRG1-I signaling in CMT1A induces a signaling axis between dedifferentiated Schwann cells, which ultimately leads to adverse glial interactions and the formation of onion bulb structures. Importantly, we show that dedifferentiation or transdifferentiation renders Schwann cells more susceptible to NRG1-I stimulation, as nerve lesion amplified onion bulb formation in NRG1-OE mice, which further underlines that the Schwann cell repair program does not reflect a simple recapitulation of nerve development[18].

Altogether, we conclude that paracrine NRG1-I signaling has evolved as a universal Schwann cell repair response. However, the ultimate function of glial NRG1-I transforms from a beneficial to a detrimental role in CMT1A disease, due to a missing counteraction (by axonal NRG1 type III) that normally restricts the temporal NRG1-I expression upon nerve repair in acute injury. Consequently, while we show that the basic cellular mechanisms are conserved between acutely and chronically injured Schwann cells, matching emerging treatment approaches have to adopt antithetic therapeutic strategies. Specifically, the pharmacological blockade of ErbB2 receptors in adult CMT1A patients might prevent aberrant glial NRG1 signaling and constitute a promising approach for future experimental trials. Indeed, ErbB2 receptor inhibitors are used in patients for breast cancer therapy and proved safe in routine clinical application[50]. In summary, glial NRG1-I/ErbB2-mediated paracrine and autocrine stimulation represents a novel neurological disease mechanism that constitutes an attractive target for therapeutic approaches of demyelinating peripheral nerve diseases.

## Methods

**Human material**. Human samples were selected according to the histological findings in sural nerve biopsies. Sural nerve biopsies with no major pathological findings (with respect to inflammation, axonal and myelin pathology, vascular pathology) were assigned to the control group. Sural nerve biopsies with a neuropathological diagnosis of a demyelinating neuropathy (with signs of demyelination, remyelination, and onion bulb formations but no major acute axonal pathology) were assigned to the experimental group of demyelinating polyneuropathies. Selection of patients was performed upon availability of fresh frozen material and according to exclusion/inclusion criteria. Samples were anonymized and processed in a blinded manner. Selected patients were of mixed age, between >30 years and <70 years of age (inclusion criteria) and did not suffer from another severe neurological disorder at time point of biopsy (exclusion criteria). No other criteria besides the described characteristics were applied, and all patients did provide informed consent in accordance with the ethical compliance guidelines. The study received ethical approval by the ethic board of the University Medical Center Göttingen, Germany.

Prior to lysis for RNA extraction, fresh frozen sural nerve biopsies were incubated in RNA*later*$^{TM}$-ICE frozen tissue transition solution (ThermoFisher) for 1 week at −80 °C. After incubation, sural nerves were transferred to room temperature (RT) and all endoneurial tissue was separated from surrounding tissue. Endoneurial tissue was than further processed as described under "RNA isolation and quantitative real-time PCR (qPCR) analysis."

**Animal models**. *Pmp22* transgenic rats (Tg(Pmp22)Kan)[30] and *PMP22* transgenic mice (Tg(PMP22)C61Clh)[31], *Pmp22*+/− mice (*Pmp22tm1Ueli*)[51], *Mpz*−/− mice (*Mpztm1Msch*)[47], Mpz+/−mice (*Mpztm1Msch*)[46], mice transgenic for neuronal *Nrg1 type I* (C57BL/6-Tg(Thy1-Nrg1*I)1Kan+/−, Thy1-Nrg1(I) tg) and neuronal *Nrg1 type III* (C57BL/6-Tg(Thy1-Nrg1*III)1Kan+/−, Thy1-Nrg1(III) tg)[24], the Dhh-Cre driver line (FVB(Cg)-Tg(Dhh-cre)1Mejr/J; DhhCre)[32], Nrg1-flox (Nrg1tm3CBm; Nrg1fl/fl)[33], and SodG93A mice (B6SJL-Tg(SOD1*G93A)1Gur/J)[52] were used and genotyped. For PCR, genomic DNA from tail biopsies was isolated by incubation in modified Gitschier buffer supplemented with TritonX-100 and proteinase K for at least 2 h at 55 °C with subsequent heat inactivation of proteinase K at 90 °C for 10 min. For routine genotyping, PCR primers were used in a coamplification reaction. Primer sequences are available upon request.

For the generation of conditional *Nrg1-I* and *Nrg1-III* transgenic mice, STOP-Nrg1 transgenes were generated by PCR amplification from a mouse *Nrg1-Iß1a* or a *Nrg1-IIIß1a* cDNA (kindly provided by Dr. Cary Lai, Indiana University) and cloned into XhoI (Nrg1-I) or SpeI/XhoI restriction sites (Nrg1-III) of the β-actin-STOP-eGFP-flox cassette in a pBluescriptKS vector. Sequences of primers and transgene cassettes are available upon request. Transgene cassettes were excised from vector backbone by digestion with SpeI/AgeI (Nrg1-I) or SalI (Nrg1-III) and injected into C57Bl/6N oocytes. The transgenic lines Nrg1-I stop flox and Nrg1-III stop flox were maintained on a C57Bl/6N background.

Experimental autoimmune neuritis was induced in rats[53]. Briefly, female rats (age: 3 months, Charles River) underwent subcutaneous injections in the hind footpad of 8 mg of bovine peripheral nerve myelin[54], emulsified in 100 µl phosphate-buffered saline (PBS), and mixed with 100 µl complete Freund's adjuvant (Difco) containing 1 mg/ml heat-inactivated *Mycobacterium tuberculosis* (H37Ra). Sciatic nerves samples were taken at the disease peak (day 15).

For the induction of toxin-induced neuropathy, Oxaliplatin was administered in adult rats by intraperitoneal injection of 2.4 mg/kg oxaliplatin for 5 consecutive days every week for 3 weeks[55]. Rats were sacrificed and sciatic nerve sampled at day 21.

A model of type I diabetes mellitus with diabetic neuropathy was generated by Streptozotocin administration in mice by intraperitoneal injection of a single high dose of 200 mg/kg Streptozotocin[56]. Mice were checked for diabetic symptoms 8 weeks after injection and only mice with elevated blood glucose levels were used for analyses. Sciatic nerves were sampled 8 weeks after injection.

All animal experiments were conducted according to the Lower Saxony State regulations for animal experimentation in Germany as approved by the Niedersächsische Landesamt für Verbraucherschutz und Lebensmittelsicherheit (LAVES) and in compliance with the guidelines of the Max Planck Institute of Experimental Medicine.

Inclusion and exclusion criteria were pre-established. Animals were randomly included according to genotyping results, age, and weight in the experiments. Animals were excluded prior to experiments in case of impaired health condition or weight difference to the average group of >10%. Exclusion criteria, during or after the experiment was performed, comprise impaired health condition of single animals not attributed to genotype or experiment (according to veterinary) or weight loss >10% of the average group. No animals had to be excluded owing to illness/weight loss in all the performed animal experiments. Exclusion criteria regarding the outcome assessment were determined with an appropriate statistical test, the Grubbs' test (or ESD method) using the statistic software Graph Pad (Prism).

Animal experiments (phenotype analyses, electrophysiology, and histology) were conducted in a single blinded fashion toward the investigator. Selection of animal samples out of different experimental groups for molecular biology/histology/biochemistry was performed randomly and in a blinded fashion.

**Sciatic nerve crush.** Adult mice were anesthetized with ketamine hydrochloride/xylazine hydrochloride (100 mg/kg body weight (BW) and 8 mg/kg BW) and the sciatic nerve was exposed. The nerve was crushed at the mid-femoral level by a standardized compression with artery forceps for 40 s. The mice were sacrificed 28 days after injury and the ipsilateral sciatic nerves were dissected and processed for electron microscopy (EM) and analyzed 4 mm distal to the injury site.

**Motor performance.** All phenotype analyses were performed by the same investigator who was blinded toward genotype. Motor performance was assessed in standardized grip strength tests for hind limbs[57] and rotarod test[58]. Hind-limb grip strength was measured by supporting the forelimbs and pulling the animal's tail toward a horizontal bar connected to a gauge. The maximum force (measured in Newton) exerted onto the T-bar was recorded and the mean of 10 repeated measurements was calculated. Rotarod test was performed with the Rotarod system 3375.5 (TSE systems) on 4 consecutive days following 2 days of training. Mice were placed on a rotating rod that was accelerated from 5 to 40 rpm in 300 s. Mean holding time of five consecutive trials on each day was recorded.

**Foot print test.** To obtain the foot print length, the hind paws of the animals were painted with black ink and then placed on a small gangway (42 cm long, 10 cm wide, walls 13 cm high) on a white paper. The paper was scanned and the foot length was measured using imageJ[59]. For analysis, the average of ten foot prints (five of each side) of each mouse was calculated.

**EM and morphometry.** Nerves were removed and fixed in 4% paraformaldehyde (PFA) and 2.5% glutaraldehyde in 0.1 M PB, contrasted with osmium tetroxide, and Epon embedded. Semithin cross-sections (0.5 μm) were cut using a microtome (Leica, UC7) with a diamond knife (Histo HI 4317, Diatome). Sections were stained with Azur II-methylene blue for 1 min at 60 °C. Light microscopic images were captured with the Axio Imager Z1 (Zeiss) in ×100 magnification. Analysis was performed on total nerve cross-section. For EM of nerves, ultrathin (50–70 nm) cross-sections were stained with 1% uranylacetate solution and lead citrate[60] and analyzed using a Zeiss EM900 or LEO912. For quantification of histological parameters as the number of unmyelinated fibers, Schwann cell nuclei, and the *g*-ratio (calculated by determination of the axon area and the area of the fibre including the myelin sheath and subsequent calculation of the diameter of a hypothetical circular area, and subsequent division of the axon diameter by the fibre diameter), at least 20 random taken pictures were captured at a magnification of ×3000.

**Internodal length-teased fibers.** Tibial nerves of 4-month-old mice and sciatic nerves of 18-day-old rats were dissected and fixed in 3% glutaraldehyde overnight at 4 °C. Nerves were washed twice in PBS and subsequently fixed in 1% Osmiumtetroxide for 2 h at RT, washed, and incubated over night in 60% glycerol at 4 °C. Nerves were then transferred in 100% glycerol and teased into single nerve fibers. For each animal, at most 40 teased fibers and 35–110 internodes were measured and analyzed. Images were taken with an Axio Imager Z1 with ×20 magnification and intermodal length was measured using ImageJ.

**Immunohistochemistry.** For immunohistochemical analysis, sciatic nerve and tibial nerve of mice was dissected and either fixed in 4% PFA and embedded in paraffin or fresh frozen in OCT. Paraffin blocks were sectioned at 5 μm and fresh frozen samples at 14 μm and mounted on Superfrost Plus slides (Thermo Scientific). Paraffin samples were deparaffinized using a standard xylol and ethanol series. Target retrieval was induced by heating in citrate buffer, samples were blocked with goat or horse serum, and incubated overnight with primary antibody against NRG1 (C-terminal, polyclonal rabbit, 1:200, Santa Cruz, #SC348), cJUN (monoclonal rabbit, 1:800, Cell Signaling, #9165), and MPZ (monoclonal mouse, 1:500, Archelos). Samples were further incubated with their corresponding secondary cyanine dyes (1:1000, Dianova: anti-rabbit, Cy3, cat.#111-165-144; anti-mouse, Cy2, cat.#115-225-071) for 1 h at RT. Cryosections were postfixed in 4% PFA for 5 min, incubated in ice-cold methanol/acetone (95:5) for 10 min, and

blocked in goat serum. The primary antibody for SOX10 (polyclonal rabbit, 1:100, Abcam, #ab107532) was incubated over three nights at 4 °C and the corresponding secondary cyanine dye (1:1000, Dianova: anti-rabbit, Cy3, cat.#111-165-144) was incubated for 1 h at RT. Nuclei were counterstained with DAPI and samples were mounted in Aqua Polymount and complete cross-sections were analyzed. Digital images of stained sections were obtained by fluorescent microscopy with the Axio Observer Z1 (Zeiss) and Zen Imaging Software (Zeiss). Images were processed by using the NIH ImageJ, Photoshop CS (Adobe), and Illustrator 10 (Adobe) software.

**BaseScope™ in situ hybridization.** In situ hybridization was performed with the BaseScope™ Detection Reagent Kit—RED (ACDbio) according to the manufacturer's instructions on paraffin-embedded samples of sciatic nerves of wild-type and PMP22 transgenic mice at the age of P18. In short, sciatic nerves of P18 mice were dissected, fixed in 4% PFA for about 20 h, and embedded in paraffin. They were cut in 5-μm sections and dried at least overnight at RT. Sections were baked at 65 °C for 1 h, deparaffinized with xylene and ethanol, and incubated with hydrogen peroxide with subsequent target retrieval for 15 min in provided target retrieval agent. Samples were incubated with provided proteaseIII for 30 min at 40 °C in HybEZ™ (ACDbio) oven followed by hybridization for 2 h at 40 °C with target and control probes and subsequent incubation with amplification reagents. The exon junction spanning probe targeted region 834–866 on gene XM_017312638.1 (5′-CCGCCGAAGGCGACCCGAGCCCAGCATTGCCTC-3′). For myelin counterstaining, the samples were washed twice in PBS after BaseScope incubation and blocked with goat serum. Samples were incubated overnight at 4 °C with MBP antibody (1:500, mM, Covance, SMI-94) and with its corresponding cyanine dye (1:1000, Dianova, anti-mouse, Cy2, cat.#115-225-071) for 1 h at RT and mounted in Aqua Polymount. Images were obtained by phase and fluorescent light microscopy (Axio Observer Z1 and Axio Imager Z1, Zeiss).

**RNA preparation and quantitative real-time PCR (qPCR) analysis.** Total RNA was extracted from nerve tissue using Qiazol Reagent and RNA from cell culture was purified using RLT lysis buffer, each according to the manufacturer's instruction (Qiagen). The integrity of purified RNA was confirmed with the Agilent 2100 Bioanalyser (Agilent Technologies). For reverse transcriptase-PCR analysis, cDNA was synthesized from total RNA using poly-Thymin and random nonamer primers and Superscript III RNase H reverse transcriptase (Invitrogen). qPCR was carried out using the Roche LC480 Detection System and GoTaq qPCR Master Mix (Promega). Reactions were carried out in four replicates. The relative quantity (RQ) of RNA was calculated using the LC480 Software (Roche). Results were depicted as histograms (generated by Microsoft-Excel 2010) of normalized RQ values, with mean RQ value in the given control group normalized to 1. Unless indicated otherwise, the housekeeping genes peptidylprolyl isomerase A (*Ppia*) and ribosomal protein, large, P0 (*Rplp0*) were used as internal standards. For PCR primer sequences, see Supplementary Table 1.

**Protein analysis.** Sciatic nerves and cultured cells were transferred to sucrose lysis buffer (320 mM sucrose, 10 mM Tris, 1 mM NaHCO$_3$, 1 mM MgCl$_2$) containing protease (Complete, Roche) and phosphatase (PhosSTOP, Roche) inhibitors and homogenized using the Precellys homogenizer (Peqlab). The homogenate was dissolved in Laemmli loading buffer (10 mM Tris, 1.55% w/v dithiothreitol (DTT), 0.6% v/v glycerine, 0.2% w/v sodium dodecyl sulfate (SDS), 0.008% w/v bromophenol blue) and 10% of a 250 mM DTT solution was added to the mixture following a short centrifugation. For all western blots, 10 μg protein (except pErbB2, 30 μg) were loaded on a 4–12% Bis-Tris gradient gel (NuPAGE, Thermo Fisher) in 3-(N-morpholino)propanesulfonic acid (MOPS) running buffer (25 mM MOPS, 25 mM Tris, 0.1% w/v SDS, 0.03% w/v EDTA) using PageRuler Plus Prestained Protein Ladder (10–250 kDa, Thermo Fisher) as loading and size control. Following separation, the proteins were transferred on an Amersham Hybond PVDF membrane (pore size 0.45 μm, GE Helathcare) with the wet blot Mini Trans-Blot Cell system (Bio-Rad) for 3 h at 100 V and 4 °C. For detection of the total protein content on the membrane, a whole protein stain was performed. The membrane was washed 5 times for 5 min each in bi-distilled water to remove any buffer residues after the transfer. Following an incubation with fluorescent Fast Green (Sigma Aldrich) staining solution (0.00084% w/v Fast Green, 30% v/v methanol, 6.7% glacial acetic acid) for 5 min at RT and protected from light. Fast Green dye bound specifically to the whole protein content on the membrane. Removal of excessive staining solution was performed by two washing steps with washing solution (30% v/v methanol, 6.7% v/v glacial acetic acid) for 30 s each at RT and protected from light. Fluorescent Fast Green signal was detected on an Intas ECL Chemostar (INTAS Science Imaging Instruments) with exposure times from 1 to 5 s, an emission wavelength of 650 nm, and a band-elimination filter with a wavelength of 665 nm. Subsequently, the membrane was washed twice in bi-distilled water for 5 min at RT after which blocking of the membrane was conducted by incubation in bovine serum albumin (BSA)-TBST solution (5% w/v BSA, 25 mM Tris, 75 mM NaCl, 5% v/v Tween 20) for 2 h at RT. Detection of immunolabeled proteins was performed using chemiluminescence (NEN Life Science Products). Western blots were incubated overnight with primary antibodies against p-AKT (Ser473, #3787), AKT (#9275), p-ERK1/2 (Thr202/Tyr204, #9101), ERK1/2 (#4695) (all pRb; 1:1000, Cell Signalling), p-ERBB2 (Y877, pRb; 1:500, Cell

Signaling, cat.#2241), ERBB2 (pRb, 1:500, Santa Cruz, sc-284), GAPDH (mM; 1:5000, Enzo Life Sciences, Clone 1D4, ADI-CSA-335-E), Actin (1:1000, EMD Millipore, Clone C4, MAB1501), NRG1-N-terminal (1:200, mM, Santa Cruz, #SC393006), and NRG1-C-terminal (1:200, pRb, Santa Cruz, #SC348).

**Phospho explorer assay**. For protein phosphorylation screening, sciatic nerves of five P30 animals for each of the three groups ($Nrg1^{fl/fl}::Dhh^{Cre}$, $PMP22$ tg::$Nrg1^{fl/fl}$, $PMP22$ tg::$Nrg1^{fl/fl}::Dhh^{Cre}$) were dissected, pooled, and applied to the Explorer Antibody Microarray (Full Moon BioSystems, Inc., Sunnyvale, CA, USA) according to the manufacturer's instructions. In short, protein was extracted from the tissue and the quality and quantity was measured with A280 spectrum. Subsequently, the proteins were biotinylated, conjugated to the array, and detected by a streptavidin dye. Phosphoarrays were imaged using a G2505B DNA microarray scanner (Agilent) at 5-µm resolution. Scan images were modified with Photoshop (Adobe): tonal values were set to 0–30 and regions containing no spots were cropped. Spot intensities were quantified manually using Delta2D (Decodon). For each spot position, a circular spot region was generated, labeled, and total signal intensity was determined. Intensity values were exported. Spot quality was evaluated for signal intensity being of above the average intensity of the internal negative control spots for at least 3 of the 4 replicate spots per epitope on each array. The arithmetic mean of the detected signals from two spatially separated spots per antibody/target was formed and used to calculate the fold change in signal intensity between the following groups: $Nrg1^{fl/fl}::Dhh^{Cre}$ versus $PMP22$ tg:: $Nrg1^{fl/fl}::Dhh^{Cre}$ (comparison 1) and $PMP22$ tg::$Nrg1^{fl/fl}$ versus $PMP22$ tg:: $Nrg1^{fl/fl}::Dhh^{Cre}$ (comparison 2). Because the data included multiple phospho-sites per protein and the Ingenuity Pathway Analysis (IPA) software (Qiagen) allows only one identifier (Swiss Prot) per target and group, all entries were curated. Briefly, the phospho-sites of one protein were first sorted by their ability to activate or inhibit the function of the corresponding protein or the protein-related activity using most recent literature concerning the regulatory ability of each phospho-site and taking the PhosphoSitePlus database (Cell Signaling). Next, a scoring scheme was applied to evaluate whether the respective phospho-site is differentially regulated between the two comparisons by the means of being upregulated in $PMP22$ tg animals (comparison 1) and downregulated in $PMP22$ tg after ablation of glial Nrg1-I (comparison 2). Therefore, all phospho-sites gathered one point each if the fold change in group 1 was ≥1.5-fold increased and ≥1.5-fold decreased in group 2, ending up in a maximum score of 2. The phospho-site with the highest score for each protein was then used for further analysis. If more than one site had the same score, the fold changes of the three groups were multiplied (a fold change <1 was reverted by calculating 1/fold change) and the phospho-site with the highest product was chosen as the most differentially regulated for the respective protein. After each protein was represented by one entry per group, fold changes of inhibitory phospho-sites were reverted (1/fold change) and the data were uploaded as log2 ratio into the IPA software using the Swiss-Prot ID as identifier. A core analysis for each comparison was performed with a fold-change cutoff of 1.5 and a $p$ value cutoff of 0.05 (Fisher's Exact Test). Next, a comparative analysis was used to identify differentially regulated pathways between the two comparisons that was performed with the recommended settings of the software. Finally, the results were refined by setting a $p$ value cutoff of –log10(45) and sorted by the most differentially, inversely regulated pathways between the 2 groups and $Z$-score.

**Primary cell culture**. Mouse and rat Schwann cells were prepared from sciatic nerves of newborn mice and rats (3–4 days old)[61]. For cell expansion, media for mouse Schwann cells were supplied with 10% deactivated Hyclone fetal calf serum (FCS; GE Healthcare & Life Technologies), 10 ng per ml medium recombinant human Neuregulin-1 beta EGF like domain (EGF-ld, Reprokine), and 4 µM forskolin (Sigma); for rat Schwann cells with 10% FCS, 100 µg/mL bovine pituitary extract (Sigma), and 4 µM Forskolin (Sigma). Experiments of directed migration were performed with the "CytoSelect™ 24-well cell migration assay 8 µm" (Boyden chamber, Cell Biolabs) according to the manufacturer's instructions. Mouse Schwann cells were grown in Dulbecco's modified Eagle's medium (DMEM) supplemented with 10% FCS, 4 µM FSK, and 10 ng/mL EGF. Cells in their second passage were seeded in PLL-coated wells (bottom chamber) and baskets (top chamber) of the migration assay (20,000 cells each). The top compartment was always seeded with wild-type Schwann cells, whereas in the bottom compartment either wild-type or $Nrg1^{fl/fl}::Dhh^{Cre}$ Schwann cells were seeded. The cells were allowed to attach for 4 h in DMEM supplemented only with 0.5% FCS and afterwards washed with PBS to remove FCS completely. The basket was then transferred to the bottom chamber and incubated for 19 h at 37 °C and 5% $CO_2$ in DMEM only or supplemented with recombinant NRG1 in rescue experiments. Migrated cells were dissociated from the membrane with detachment buffer, lysed, and quantified using CyQuantGR Fluorescent Dye[62] and a fluorescent plate reader at 480 nm/520 nm (Synergy MX, Biotec). The experiment was conducted with three independent cell culture preparations.

Survival experiments were performed in three biological replicates, respectively, with rat wild-type Schwann cells and mouse $Nrg1^{fl/fl}$ and $Nrg1^{fl/fl}::Dhh^{Cre}$ Schwann cells. Seeded cells were washed five times in DMEM and kept on DMEM only. Rat wild-type Schwann cells were treated with an ErbB2-Inhibitor (CAS 928207-02-7, Calbiochem, 30 µM in dimethyl sulfoxide (DMSO)) or DMSO as control (1:1000). Images were taken at 0 and 48 h with the Axio Observer Z1 (Zeiss). Mutant mouse

Schwann cells were kept in DMEM for 72 h and analyzed for apoptosis as described below.

Dorsal root ganglia (DRGs) cultures were prepared by isolating DRGs from wild-type or $PMP22$ transgenic[31] mouse embryos at embryonic day 13.5 (E13.5) and plated according to the standard procedure[63]. DRGs were dissociated with trypsin and plated at a density of $1 \times 10^5$ cells per 1 mm coverslip coated with collagen (Gibco). The cells were kept in growth medium comprised of minimum essential medium (Gibco) supplied with 10% deactivated Hyclone fetal bovine serum (GE Healthcare & Life Technologies) and 50 ng/ml NGF (Alomone laboratories) for 1 week. In order to induce myelination, growth medium was supplemented with 50 ng/ml ascorbic acid (Sigma) every other day.

**Schwann cell apoptosis and proliferation**. For analysis of Schwann cell proliferation, mice were injected with 40 mg/kg BW EdU (Invitrogen) intraperitoneally at the age of P10. Sciatic nerves were dissected after 24 h, fixed in 4% PFA, and embedded in paraffin. Cross-sections were stained with the "Click-iT EdU Imaging Kit Alexa Fluor555" (Invitrogen) according to the manufacturer's instructions and counterstained with MBP (1:1000, mM, Covance # SMI-94).

For analysis of apoptosis, sciatic nerves of 11-day-old mice were dissected (i.e., in the phase of physiological postnatal Schwann cell elimination). Nerves were fixed in 4% PFA, embedded in paraffin, and longitudinal sections (14 µm thickness) were mounted on Superfrost Plus slides.

Primary mouse Schwann cells were prepared and cultured for a survival assay as described above. Apoptosis was assessed by conducting a TUNEL assay with the "DeadEnd™ Colorimetric TUNEL System" Kit (Promega) according to the manufacturer´s instructions. The apoptosis rate was quantified by counting the number of TUNEL-positive nuclei over the total number of nuclei.

**Electrophysiology**. For standard electroneurography[29], mice were anesthetized with ketamine hydrochloride/xylazine hydrochloride (100 mg/kg BW/ 8 mg/kg BW) and pairs of steel needle electrodes (Schuler Medizintechnik, Freiburg, Germany) were placed subcutaneously along the nerve at the sciatic notch (proximal stimulation) and the tibial nerve above the ankle (distal stimulation), respectively. Supramaximal square wave pulses lasting 100 ms were delivered using a Toennies Neuroscreen® (Jaeger, Hoechsberg, Germany). CMAP was recorded from the intrinsic foot muscles using steel electrodes. Both amplitudes and latencies of CMAP were determined. The distance between the two sites of stimulation was measured alongside the skin surface with fully extended legs and NCVs were calculated automatically from sciatic nerve latency measurements.

**Statistical analysis**. For power analysis, the software G*Power Version 3.1.7. was used. Power analyses were performed before conducting in vivo experiments (a priori). Adequate power (1 − beta error) was defined as ≥80% and the alpha error as 5%.

Data are expressed in mean ± standard deviation (SD) unless indicated otherwise. All other data was processed and statistically analyzed using MS Excel and GraphPad Prism v6.04, unless indicated otherwise. The statistical test that was used to analyze the data is indicated in the figure legends. Briefly, for comparing two groups Student's $T$ test was used, for comparing more than two groups one-way analysis of variance (ANOVA) with appropriate post test was used, and for comparing two or more groups for more than one time point (longitudinal) two-way ANOVA with appropriate post test was used. Statistical differences were considered to be significant when $p < 0.05$ (*$p < 0.05$, **$p < 0.01$, ***$p < 0.001$).

**Reporting Summary**. Further information on experimental design is available in the Nature Research Reporting Summary linked to this article.

## Data availability

All relevant data of the present manuscript are available from the corresponding authors on reasonable request. The source data underlying Figs. 1a–d, g, h, 2a, b, d–i, k, 3c–h, k, 4a–g, 5a–f, and 6a–i are available as a Source Data file.

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

## Acknowledgements

We thank C. Huxley (Imperial College School of Medicine, London) for providing *Pmp22* transgenic mice, C. Birchmeier (MDC Berlin) for providing *Nrg1$^{flox}$* mice, R. Martini (U Wuerzburg) and Melitta Schachner (ZMNH, Hamburg) for providing *Mpz$^{-/-}$* mice, and Dies Meijer (U Edinburgh) for *Dhh$^{Cre}$* animals. We are grateful to A. Mott, T. Durkaya, C. Maack, T. Ruhwedel, A. Fahrenholz, T. Pawelz, and M. Wehe (MPI of Experimental Medicine) as well as to A. Wohltmann (Institute of Neuropathology, Göttingen), S. Richter (Department of Neuropathology, Leipzig), and J. Craatz (Institute of Anatomy, Leipzig) for technical support. This work was funded by the ERA-NET for Research Programs on Rare Diseases E-RARE-3 (01GM1605) and the German Ministry of Education and Research (BMBF, FKZ: 01GM1511C). M.C.S.-B. acknowledges funding by Doctoral Fellowships from the DAAD, the Institute of Technology of Costa Rica (TEC) and the Ministry of Science, Technology, and Telecommunication of Costa Rica (MICITT). M.H.S. holds a Heisenberg Fellowship from the Deutsche Forschungsgemeinschaft (DFG) and acknowledges funding by a DFG research grant (SCHW741/4-1). K.-A.N. is supported by the DFG (CNMPB and SPP1757) and holds an ERC Advanced grant.

## Author contributions

R.F. and R.M.S. designed and supervised the study, performed experiments and wrote the manuscript. D.A. performed and planned the experiments and contributed to the manuscript. T.A.A., V.S., D.H., E.S., M.C.S.-B., A.K., T.G., K.K., T.K., and C.F. contributed to experiments. W.M., M.K. and I.B. contributed to electron microscopy. W.B. contributed to human samples. W.C.M. contributed to immunohistochemistry. M.W.S. contributed CMT1A rat samples and contributed to discussions. M.H.S. generated and provided *Nrg1-I* and *-III$^{stop-flox}$* mice and contributed to the manuscript and discussion. K.A.N. contributed to the manuscript and discussion.

## Additional information

**Competing interests:** The authors declare no competing interests.

