## [Peer Review File · Nature Communications]

REVIEWERS' COMMENTS:

Reviewer #1 (Remarks to the Author):

In this manuscript, the authors examine the molecular mechanisms mediating the formation of onion bulbs in models of peripheral neuropathies. Using a wide range of transgenic models, the authors clearly demonstrate that onion bulb formation and correlated modulation of Schwann cell development and hypermyelination results from aberrant signaling of Schwann cell derived Neuregulin-1 type 1 (NRG1-1).

This is an interesting manuscript that contains a vast amount of data. Overall, the work is of a very high quality and the results clearly support the interpretations of the authors. The manuscript has been reworked multiple times in response to previous reviews and is considerably improved. It has, however, become a little disjointed in places and would benefit from minor revision. There are also elements of the results that are not placed into context and the reader is left uncertain about the significance. For example, in Figure 2 m the authors describe changes in the NF spacing in control and CMT1A animals but the significance of these observations is unclear. Do the authors believe the onion bulb phenotype results in axonal constriction or delayed NF transport? If it is constriction, how does that relate to changes in g-ratios? Likewise, in Fig 3g with the changes in g-ratio in the NRG1-1 OE are the axon diameters similar in the phenotypes? They look smaller in the NRG1-1 OE.

If the results are not directly relevant to the main point of the paper maybe they could be eliminated to make the paper easier to read.

Minor points.

The legend to Figure 2K (line 732) discusses arrows in a "blow up" but there is no blow up. Supplementary Figure 3 panels g and h do not seem to add useful data and could be removed.

Reviewer #2 (Remarks to the Author):

This very complete study provides compelling evidence that Schwann cell NRG1 expression, activated by reduced axonal type III expression, drives onion bulb formation by Schwann cell in a mouse model of CMT1. This study is an important contribution to our understanding of this pathological hallmark of peripheral neuropathies. The authors have addressed all of my prior concerns, including clarifying their disease model and revamping their discussion. I have no remaining issues.

Two minor phrasing corrections to consider: The authors used the phrase: proved safe rather than proved safe

"Indeed, the hypomyelination of large fibers in CMT1A has been attributed to the NCV slowing characteristic...." Think it should be other way around - the NCV slowing is attributed to hypomyelination

Point to Point Response to the Reviewers Comments:

Reviewer #1 (Remarks to the Author):

In this manuscript, the authors examine the molecular mechanisms mediating the formation of onion bulbs in models of peripheral neuropathies. Using a wide range of transgenic models, the authors clearly demonstrate that onion bulb formation and correlated modulation of Schwann cell development and hypermyelination results from aberrant signaling of Schwann cell derived Neuregulin-1 type 1 (NRG1-1). This is an interesting manuscript that contains a vast amount of data. Overall, the work is of a very high quality and the results clearly support the interpretations of the authors. The manuscript has been reworked multiple times in response to previous reviews and is considerably improved.

We thank the Reviewer for the positive judgement of our manuscript.

1) It has, however, become a little disjointed in places and would benefit from minor revision. There are also elements of the results that are not placed into context and the reader is left uncertain about the significance. For example, in Figure 2 m the authors describe changes in the NF spacing in control and CMT1A animals but the significance of these observations is unclear. Do the authors believe the onion bulb phenotype results in axonal constriction or delayed NF transport? If it is constriction, how does that relate to changes in g-ratios? Likewise, in Fig 3g with the changes in g-ratio in the NRG1-1 OE are the axon diameters similar in the phenotypes? They look smaller in the NRG1-1 OE. If the results are not directly relevant to the main point of the paper maybe they could be eliminated to make the paper easier to read.

We agree with the reviewer that the neurofilament spacing constitutes a descriptive finding and that a more detailed understanding of the underlying molecular mechanism requires future studies. As the reviewer proposed, we therefore eliminated the neurofilament spacing from the revised version of the manuscript.

Regarding the g-ratio and the axonal diameters in the NRG1-OE animals, we now analysed the axon diameters of NRG1-OE animals and could show that the axonal diameters in NRG1-OE animals are unaltered. Hence, the g-ratio changes are not secondary to alterations of the axonal diameter. We clarified this aspect in the revised version of the manuscript (see Supplementary Figure 2g and Text Line 206).

Minor points.

The legend to Figure 2K (line 732) discusses arrows in a "blow up" but there is no blow up.

We thank the reviewer for this remark and we adapted the figure legend accordingly.

Supplementary Figure 3 panels g and h do not seem to add useful data and could be removed.

We removed the respective supplementary panels as suggested.

Reviewer #2 (Remarks to the Author):

This very complete study provides compelling evidence that Schwann cell NRG1 expression, activated by reduced axonal type III expression, drives onion bulb formation by Schwann cell in a mouse model of CMT1. This study is an important contribution to our understanding of this pathological hallmark of peripheral neuropathies. The authors have addressed all of my prior concerns, including clarifying their disease model and revamping their discussion. I have no remaining issues.

We thank the reviewer for reviewing our manuscript and the positive evaluation of our study.

Two minor phrasing corrections to consider: The authors used the phrase: proved save rather than proved safe. "Indeed, the hypomyelination of large fibers in CMT1A has been attributed to the NCV slowing characteristic...." Think it should be other way around - the NCV slowing is attributed to hypomyelination

We thank the reviewer for both helpful comments and we changed the respective sentences accordingly.